# Diverse Conventions for Human-AI Collaboration

**Bidipta Sarkar**
Stanford University
bidiptas@stanford.edu

**Andy Shih**
Stanford University
andyshih@cs.stanford.edu

**Dorsa Sadigh**
Stanford University
dorsa@cs.stanford.edu

## Abstract

Conventions are crucial for strong performance in cooperative multi-agent games, because they allow players to coordinate on a shared strategy without explicit communication. Unfortunately, standard multi-agent reinforcement learning techniques, such as self-play, converge to conventions that are arbitrary and non-diverse, leading to poor generalization when interacting with new partners. In this work, we present a technique for generating diverse conventions by (1) maximizing their rewards during self-play, while (2) minimizing their rewards when playing with previously discovered conventions (cross-play), stimulating conventions to be semantically different. To ensure that learned policies act in good faith despite the adversarial optimization of cross-play, we introduce *mixed-play*, where an initial state is randomly generated by sampling self-play and cross-play transitions and the player learns to maximize the self-play reward from this initial state. We analyze the benefits of our technique on various multi-agent collaborative games, including Overcooked, and find that our technique can adapt to the conventions of humans, surpassing human-level performance when paired with real users.[1]

## 1   Introduction

In multi-agent cooperative games, understanding the intent of other players is crucial for seamless coordination. For example, imagine a game where teams of two players work together to cook meals for customers. Players have to manage the ingredients, use the stove, and deliver meals. As the team works together, they decide how tasks should be allocated among themselves so resources are used effectively. For example, player 1 could notice that player 2 tends to stay near the stove, so they instead spend more time preparing ingredients and delivering food, allowing player 2 to continue working at the stove. Through these interactions, the team creates a "convention" in the environment, which is an arbitrary solution to a recurring coordination problem [20, 3]. In the context of reinforcement learning, we can represent a specific convention as the joint policy function that each agent uses to choose their action given their own observation. If we wish to train an AI agent to work well with humans, it also needs to be aware of common conventions and be flexible to changing its strategy to adapt to human behavior [2, 29].

One approach to training an AI agent in a cooperative setting, called self-play [33], mimics the behavior of forming conventions by training a team of agents together to increase their reward in the environment, causing each player to adapt to the team's behavior. Eventually, self-play converges to an equilibrium, where each player's policy maximizes the score if their team does not change.

Unfortunately, self-play results in incredibly brittle policies that cannot work well with humans who may have different conventions [29], including conventions that are more intuitive for people to use. In the cooking task, self-play could converge to a convention that expects player 2 to bring onions to AI player 1 from the *right* side. If a human player instead decides to bring onions from the *left* side to the AI player 1, the AI's policy may not react appropriately since this is not an interaction

---

[1]Supplemental videos can be found on our website along with source code and anonymized user study data.

37th Conference on Neural Information Processing Systems (NeurIPS 2023).

that the agent has experienced under self-play. One way to address this issue is to train a *diverse* set of conventions that is representative of the space of all possible conventions. With a diverse set of conventions, we can train a single agent that will work well with arbitrary conventions or adapt to new conventions when working with humans.

To generate a diverse set of policies, prior works have used statistical divergence techniques that measure diversity as the difference in action distributions at each state between policies [9]. They assert that policies that result in different actions at each state will lead to players taking different paths in the environment, hence increasing diversity. However, it is often easy to trick this proxy objective for diversity by taking *different actions* that still lead to the same outcome. For example, in Fig. 1, we see two conventions that divide the task in the same way but differ in navigation. Even though this difference does not interfere with the partner and the conventions are perfectly compatible with one another, statistical divergence metrics treat these two policies as entirely different. Therefore, statistical diversity techniques do not fundamentally capture whether the conventions follow genuinely different strategies.

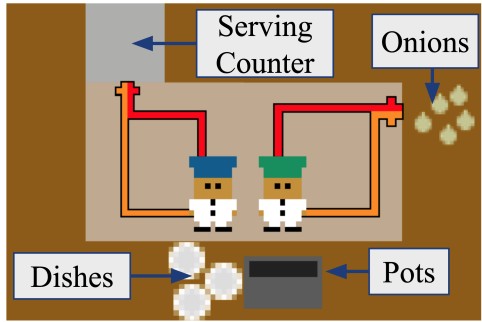

Figure 1: Example of conventions in Overcooked. Each team needs to (1) pick up onions, (2) drop them into pots, (3) fill up a dish, and (4) drop it off at the counter. There are 2 conventions shown above, a red convention and an orange convention, which only differ in terms of navigation, so they are compatible with one another.

We instead use the incompatibility of different conventions as a measure of diversity [7, 8]. Suppose we already have a set of conventions and we want to add a new convention that is different from all the conventions we have already found. We can measure the reward we would expect to receive if a player from this new convention played with another player from an existing convention, which we refer to as the cross-play reward. If this convention has a high cross-play, adding this convention would be redundant, because a policy from one of the existing conventions would already work with this new convention. On the other hand, a convention that has low cross-play with our set of conventions has a different strategy for the game and should be added to the set, because a human player might use that strategy.

Although directly minimizing cross-play rewards takes us closer to our goal of learning diverse conventions, it can result in poorly behaved conventions due to the adversarial nature of this optimization. In particular, a failure mode we refer to as *handshakes* occurs when policies intentionally sabotage the game upon realizing that the task rewards are inverted. For example, one convention could decide that players should always pick up and place down a plate at the start of the game, which is considered a handshake to establish one's identity without significantly decreasing its self-play reward. If a player 1 from this convention plays with a partner that does not execute the handshake, this player 1 can intentionally sabotage the game and achieve a low cross-play score even if the fundamental strategies are similar. To fix this issue, we introduce mixed-play, a setting where players start with a "mixed-state generation" phase in which they sample actions from self-play or cross-play, but transition into pure self-play at a random timestep with no notice. Since players receive a positive reward during self-play, handshakes cannot reliably indicate whether they should maximize their reward (self-play) or minimize it (cross-play), resulting in conventions that act in good faith.

We propose the CoMeDi algorithm, which stands for "**C**ross-play **o**ptimized, **M**ixed-play **e**nforced **Di**versity." CoMeDi generates a sequence of conventions that minimize the cross-play with the most compatible prior conventions in the sequence while using mixed-play to regularize the conventions. We evaluate our method on three environments: Blind Bandits, Balance Beam, and Overcooked [5, 34, 26]. We see that CoMeDi is able to generate diverse conventions in settings where statistical diversity and naive cross-play fail. Finally, we evaluate our generated pool of conventions in a user study by training an agent that is aware of the conventions in the pool and testing it with human users in Overcooked. We find that CoMeDi significantly outperforms the pure self-play and statistical diversity baselines in terms of score and user opinions, surpassing human-level performance.

## 2   Preliminaries

We model cooperative games as a multiagent Markov decision process [3]. The MDP, $\mathcal{M}$, is the tuple $(\mathcal{S}, \mathcal{A}, \mathcal{P}, r, \mathcal{O}, \gamma, T)$, where $\mathcal{S}$ is the (joint) state space and $\mathcal{A} = A^1 \times A^2$ is the joint action space. Although we only analyze 2-player games in the main text, we detail an extension to larger teams in Appendix A.5. The transition function, $\mathcal{P} : \mathcal{S} \times \mathcal{A} \times \mathcal{S} \to [0, 1]$, is the probability of reaching a state given a current state and joint action. The reward function, $r : \mathcal{S} \times \mathcal{A} \to \mathbb{R}$, gives a real value reward for each state transition. The observation function, $\mathcal{O} : \mathcal{S} \to O^1 \times O^2$, generates the player-specific observations from the state. Finally, $\gamma$ is the reward discount factor and $T$ is the horizon.

Player $i$ follows some policy $\pi^i(a^i \mid o^i)$. At time $t$, the MDP is at state $s_t \in \mathcal{S}$, so the agents receive the observations $(o_t^1, o_t^2) = \mathcal{O}(s_t)$ and sample an action from their policy, $a_t^i \sim \pi^i(a_t^i \mid o_t^i)$. The environment generates the next time step as $s_{t+1} \sim \mathcal{P}(s_t, a_t, s_{t+1})$ and the shared reward as $r(s_t, a_t)$. The trajectory is defined as the sequence of states and actions in the environment, which is $\tau = (s_0, a_0, \ldots s_{T-1}, a_{T-1}, s_T)$. The discounted return for a trajectory is $R(\tau) = \sum_{t=0}^{T-1} \gamma^t r(s_t, a_t)$. The expected return for a pair of agent-specific policies is $J(\pi^1, \pi^2) = \mathbb{E}_{\tau \sim (\pi^1, \pi^2)}[R(\tau)]$. For simplicity of exposition, we assume that the agents in the environment are not ordered, so $J(\pi^1, \pi^2) = J(\pi^2, \pi^1)$ for all $\pi^1, \pi^2$, although our method directly handles situations where order matters.

We refer to an instance of the joint policy of the agents $\pi = \pi^1 \times \pi^2$ as a **convention**, following the definition of a convention as an arbitrary solution to a recurring coordination problem in literature [20, 3]. When the joint policy forms a Nash equilibrium, we refer to it as an **equilibrium convention**, based on some some stricter definitions of conventions in literature [13, 19]. In equilibrium conventions, neither player can get a higher expected score with the current partner by following a different policy.

### 2.1   Problem Definition

In this paper, we study the problem of generating a diverse set $D$ of conventions that are representative of the set of potential conventions in the environment at test time, $D_{\text{test}}$. To determine how well $D$ generalizes to the test set, we can calculate a "nearest neighbor" score by finding which convention in $D$ generates the highest reward for each convention in the test set, which corresponds to $S$ in Eq. (1):

$$S(D) = \mathbb{E}_{\pi \in D_{\text{test}}} [\max_{\pi^* \in D} J(\pi^*, \pi)]. \tag{1}$$

In practice, we can also treat our generated set $D$ as the set of *training* policies used to learn a generalizable agent [21] for zero-shot coordination. We rely on off-the-shelf multitask training routines [6] for learning the generalizable agent, which have shown to be effective in learning an agent that can coordinate with new *test* partners [27, 21]. In particular, given the set $D$ of diverse conventions, we can train a convention-aware agent using behavior-cloning [22, 17] in a similar fashion to QDagger [1]:

$$\mathcal{L}(\hat{\pi}, D) = -\mathcal{J}(\hat{\pi}, \hat{\pi}) - \frac{\lambda}{|D|} \sum_{\pi \in D} \mathbb{E}_{(o,a) \sim \pi} [\log(\hat{\pi}(a|o))], \tag{2}$$

where $(o, a) \sim \pi$ represents the distribution of observation-action pairs sampled from self-play rollouts of $\pi$, and $\lambda$ is a tunable hyperparameter for the weight of the BC component. At evaluation time, we pair the convention-aware agent $\hat{\pi}$ with a new *test* partner and measure the task reward:

$$\mathcal{J}(\hat{\pi}, D_{\text{test}}) = \mathbb{E}_{\pi \in D_{\text{test}}} [\mathcal{J}(\hat{\pi}, \pi)]. \tag{3}$$

For both Eq. (1) and Eq. (3), the test agent distribution $D_{\text{test}}$ may be too expensive to sample from during the training loop, e.g., when this test dataset is based on samples from human users. Instead, we would like to generate a high quality set of *training* agents that is representative of the space of $D_{\text{test}}$. Intuitively, a more diverse set $D$ will improve the generalization of the convention-aware $\hat{\pi}$ trained on the multi-task objective in Eq. (2).

A common recipe for generating a diverse set $D$ is to train individual agents via standard self-play, and propose a diversity regularization loss that encourages the set $D$ to be more "diverse" with respect to the regularization term:

$$\mathcal{L}(D) = -\sum_{\pi \in D} [\mathcal{J}(\pi, \pi)] + \alpha \, \mathcal{C}(D). \tag{4}$$

In Eq. (4), the overall loss of the set $D$ is a combination of the diversity regularization $\mathcal{C}$ (scaled by $\alpha$) and the self-play rewards of the individual agents in the set $D$. The choice of regularization term $\mathcal{C}$ requires a difficult balance between tractable computation, alignment to the downstream objective (Eq. (1), Eq. (3)), and ease of optimization of Eq. (4). In Section 2.2 and Section 2.4, we highlight some existing diversity metrics and their limitations.

## 2.2 Statistical Diversity

The majority of prior works on diversity in deep reinforcement learning are based on the principle of maximizing statistical divergence [11, 10, 21, 9, 15]. In the realm of zero-shot coordination, Trajectory Diversity (TrajeDi) [21] uses the Jensen-Shannon divergence between the trajectories induced by policies as a diversity metric. The method of learning adaptable agent policies (ADAP) [9] maximizes the KL-divergence between the action distributions of two policies over all states. We compare against ADAP as a baseline by adding its diversity term in Eq. (5) to the MAPPO algorithm [36]:

$$\mathcal{C}_{\text{ADAP}}(D) = \mathbb{E}_{s \in S} \left[ \mathbb{E}_{\substack{\pi_1, \pi_2 \in D \\ \pi_1 \neq \pi_2}} \exp(-D_{\text{KL}}\left(\pi_1(s) || \pi_2(s)\right)) \right]. \tag{5}$$

While statistical divergence techniques are computationally simple, they do not use the task reward and thus can be poorly aligned with the downstream objectives in Eq. (1) and Eq. (3). For example, consider the introductory Overcooked example in Fig. 1. There are many possible routes that the players can take to complete their individual task, leading to distinct but semantically similar trajectories. Ideally, we want a diversity regularizer to penalize trivial variations that are irrelevant to the task at hand. Unfortunately, statistical diversity would fail to recognize these semantic similarities.

## 2.3 Cross-Play Minimization Diversity

Recent work on generating strong zero-shot agents in cooperative games have also used the idea of generating a diverse set of conventions by minimizing cross-play. The LIPO algorithm [7] follows a similar approach to our baseline cross-play minimization algorithm described in Section 3.1, but it does not fundamentally tackle the issue of handshakes described in Section 3.2. The ADVERSITY algorithm [8] also follows an approach of minimizing cross-play scores, but addresses the issue of handshakes in Hanabi through a belief reinterpretation process using the fact that it is a partially observable game. Unfortunately, the ADVERSITY algorithm would not function in our fully observable settings since there is no hidden state for belief reinterpretation. We provide a more detailed comparison of our approach to these prior works in Appendix C.

## 2.4 Other Approaches to Diversity

Other prior works attempt to modify the training domain to induce a variety of conventions [32, 35, 23]. A common aspect to modify is the reward, which can be randomized [32] or shaped by having an external agent try to motivate a certain behavior [35]. This requires extra effort from an environment designer to specify the space of the reward function, which may be impractical in complex environments. Additional approaches using population-based training modify pre-specified aspects of the environment to generate a set of agents [16], or generate agents using different random seeds but require a human to manually select diverse agents [31]. Another set of work focuses on the easier setting of using imitation learning to learn to play with humans [18, 30, 28], but this assumes access to potentially expensive human behavioral data.

We do not explicitly compare CoMeDi to these other approaches in the main text because they assume access to the internal mechanisms of the environment or require external data, so they are not drop-in replacements for self-play, unlike CoMeDi or statistical diversity techniques.

## 3 Method

Given the drawbacks of current statistical diversity methods at penalizing semantically similar trajectories, we seek alternative diversity regularization terms that can recognize semantically diverse behavior from trivial variations in trajectory space. A good mechanism for identifying trivial variations

is through task reward: if swapping the behavior of two agents does not affect the attained task reward, then the differences between their behavior are likely irrelevant to the task at hand. For example, consider the effect of the blue player following the red trajectory while the green player follows the orange trajectory in Fig. 1. The swapped teams will still attain the same high reward as the players following the same color trajectories, suggesting that these behaviors are not semantically different.

With this insight, we look to measure cross-play, the task reward attained when pairing together players that were trained separately. We propose a diversity regularization based on minimizing cross-play reward, and describe its benefits compared to statistical diversity. However, pure cross-play faces an adversarial optimization challenge that we refer to as *handshakes*, making it difficult to optimize. In Section 3.3, we describe mixed-play, our novel technique for mitigating handshakes in cross-play while still retaining the benefits of reward-aligned diversity metrics.

## 3.1 Cross-Play Minimization

The idea of cross-play is simple: pair two agents from different conventions together and evaluate their reward on the task. If those two conventions are semantically different, then the cross-play reward should be low. On the other hand, if the conventions of the two agents are similar, then their cross-play reward should be high, and it should be penalized by our diversity regularization term.

We can also derive the notion of cross play minimization by trying to *maximize* the nearest neighbors score $S(D)$ from Eq. (1). Suppose we already have found $n-1$ conventions in the set $D_{n-1}$, and we wish to add a new convention $\pi_n$ to construct the new set $D_n$. We can evaluate a lower bound for the improvement of $S$ as

$$S(D_n) \geq S(D_{n-1}) + p(\pi_n)(J(\pi_n, \pi_n) - \max_{\pi^* \in D_{n-1}} J(\pi_n, \pi^*)), \qquad (6)$$

where $p(\pi_n)$ is the probability of choosing policy $\pi_n$ at test-time. We explore the algorithmic implications of $p(\pi_n)$ in Section 3.2.

Using the result above, we extend cross-play minimization to a set of $n$ agents by using the cross-play reward between each convention and the most compatible existing convention as the regularization term $\mathcal{C}_X$ in Eq. (7). To determine which existing convention is the most compatible, we can simply collect cross-play buffers with all existing conventions and choose the convention with the highest expected return. Note that $\mathcal{C}_X$ is minimized, so rewards are *inverted* under cross-play:

$$\mathcal{C}_X(D) = \sum_{i=2}^{n} \max_{\pi^* \in D_{i-1}} J(\pi_i, \pi^*). \qquad (7)$$

If we integrate Eq. (7) into the full loss function from Eq. (4), the cross-play loss is simply equal to the sum of the lower-bound expressions in Eq. (6) for all conventions in $D$ when $p(\pi_i)$ is ignored.

## 3.2 Handshakes: Challenges of Optimizing Cross-Play

Recall that the downstream objective is to maximize the nearest neighbors score from Eq. (1), meaning that we want policies that work well with the unseen test distribution. Unfortunately, directly minimizing cross-play can lead to undesirable policies for the downstream objective. In particular, policies optimized with pure cross-play often take illogical actions that would actively harm the task reward when they are in a cross-play setting while still performing well in self-play.

This sabotaging behavior is also observed in concurrent work in Hanabi [8]. They specifically analyze cases where players intentionally use unplayable cards to quickly end the game. Their "self-play worst response" policy, analogous to our pure cross-play minimization strategy for $\pi_2$, intentionally sabotages the game under cross-play, resulting in a brittle policy that identifies when it is under cross-play. To resolve this issue in Hanabi, ADVERSITY uses belief reinterpretation, similar to Off-Belief Learning [14], in order to construct a fictitious trajectory consistent with the observed trajectory. This prevents sabotaging behavior because observations are equally likely to come from self-play or cross-play. Although this technique is very effective in partially observable environments, like Hanabi, this unfortunately does not solve the issue in our fully observable settings because there is no hidden state that can be reinterpreted to construct plausible self-play trajectories given a cross-play trajectory.

We hypothesize that the challenge with the optimization of cross-play regularization arises when agents can easily discriminate between cross-play and self-play, as illustrated in Fig. 2. For example, consider the earlier Overcooked example in Fig. 1, where we now pair the red convention for the blue player and orange convention for the green player during cross-play. As we repeatedly update the policies based on minimizing cross-play, the agents will learn to recognize when they are being evaluated under cross-play and deliberately sabotage the game. The blue player can learn to develop a handshake, like shown in Fig. 2, and sabotage the game based on the green player's reaction. Note that the strategy after the *handshake* may be similar across both conventions, but they will still

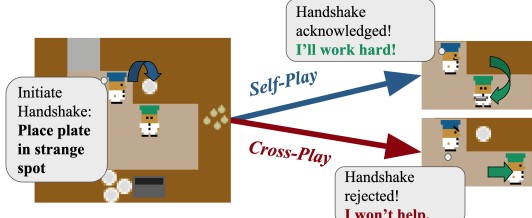

Figure 2: Example of a handshake in Overcooked. The blue player initiates a handshake by placing a plate in a strange location. Under self-play, the green player will pick up the plate, but otherwise it will continue picking up onions. If the blue player sees that the plate has not moved, it will sabotage the game.

have a low cross-play due to intentional sabotage. As a result, these agents have fooled our cross-play diversity metric into treating their behaviors as semantically different.

More generally, this problem occurs when there is a mismatch in observation distributions (or history distribution for recurrent policies) between self-play and cross-play, which allows the agents to infer the identity of their partner and the inversion of rewards. In fact, a mismatch can be identified as early as the second timestep: in the first timestep, both agents perform a *handshake* – an action intended to reveal the identity of the players. If the handshakes match, then the two agents cooperate to solve the task. Otherwise, they sabotage the game, e.g., by doing nothing for the rest of the episode.

Since a strong self-play policy is under-specified at many states, all of the mechanisms of handshake signalling and sabotaging can be encoded into a policy with almost no reduction in self-play performance. We would not expect this type of behavior from a convention in the test set, so this policy, $\pi$, would have a very low $p(\pi)$ in Eq. (6) relative to other trained policies. The outcome of a cross-play minimization procedure (Eq. (7)) plagued by handshakes is a set of agents who have high individual self-play, low pairwise cross-play, yet may still be using semantically similar conventions.

### 3.3 Mixed-Play

To mitigate the issues of handshakes, we propose *mixed-play*, which improves cross-play regularization by expanding the valid distribution of "self-play states" and reducing under-specification. Mixed-play rolls out episodes using varying combinations of self-play and cross-play to enlarge state coverage. All the states in the coverage will be associated with the positive rewards of self-play, so agents cannot infer reward inversion and sabotage the game. For example, if the cross-play situation from Fig. 2 becomes a valid state that self-play can continue

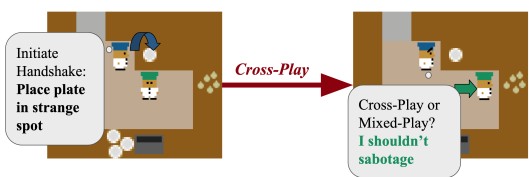

Figure 3: Visualization of how mixed-play resolves handshakes. Agents must act in good faith since they might switch to self-play.

from, the blue agent can no longer confidently sabotage the game when it encounters this state, since this harms the self-play score as well.

Mixed-play consists of two phases in each episode: mixed-state generation and self-play. We choose a random timestep within the episode that represents the length of the first phase. Until this timestep occurs, we randomly sample the action from self-play or cross-play for both players, but do not store these transitions in the buffer. For the rest of the episode, we perform the second phase: taking self-play actions and storing them in the buffer. When optimizing, we treat this new buffer the same as we would treat self-play, but modified with a positive weight hyperparameter, $\beta$, representing the importance of mixed-play. Unlike ADVERSITY, our technique does not assume that the environment is partially observable and does not require privileged access to the simulator, making it applicable to our fully observable settings.

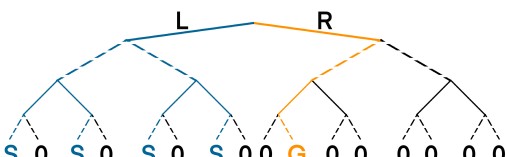

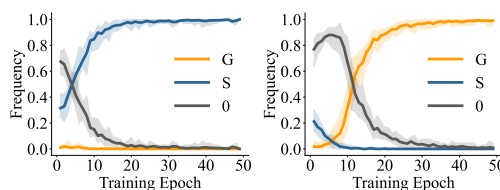

Figure 4: The Blind Bandits environment with $k = 2$ steps. The solid lines indicate the first player's actions while the dashed lines indicate the second player's actions. The blue line represents the conventions that converge to the $S$ reward while the orange line represents the convention that converge to the $G$ reward.

Figure 5: Frequency of self-play scores during the optimization of the two conventions using CoMeDi. The left plot is the first convention while the right plot is the second convention. We choose $k = 3$ as the steps in the environment, and the shaded region indicates the frequency bounds from 10 independent initializations.

After training with mixed-play, the agent cannot determine the current identity of the partner. In particular, even if the partner fails the handshake in an earlier timestep, the setting might be mixed-play instead of cross-play, as visualized in Fig. 3. If $\beta$ is large enough, the agent will learn that handshakes significantly harm the scores in the second phase of mixed-play and will learn to act in good faith at all timesteps. In the following section, we present our full algorithm that incorporates mixed-play into our technique for generating diverse conventions.

### 3.4 CoMeDi: Self-play, Cross-play, and Mixed-play

So far, we have described how cross-play is useful for constructing a reward-aligned diversity metric, while mixed-play ensures that trained agents always act in good faith. In the full training regime, which we name CoMeDi, we (1) maximize self-play rewards to ensure that agents perform well in the environment, (2) minimize cross-play rewards with previously discovered conventions to ensure diversity, and (3) maximize rewards from the collected mixed-play buffers to ensure that conventions act in good faith. Note that we collect separate buffers for self-play, cross-play, and mixed-play to estimate their respective expected returns.

To train convention $\pi_n$, the full loss function we evaluate is

$$\mathcal{L}(\pi_n) = -\mathcal{J}(\pi_n, \pi_n) + \alpha \mathcal{J}(\pi_n, \pi^*) - \beta \mathcal{J}_\mathrm{M}(\pi_n, \pi^*), \tag{8}$$

where $\mathcal{J}_\mathrm{M}$ represents the expected mixed-play reward, and $\pi^*$ is the most compatible convention to $\pi^n$ found in $D_{n-1}$. We also have tunable parameters $\alpha$ and $\beta$ representing the weight of the cross-play and mixed-play terms. Pseudocode and specific implementation details for the algorithm incorporating this loss, including the mixed-play buffer generation, is presented in Appendix A.

## 4 Experiments

We now describe our experiments and results using three different environments: a Blind Bandits environment, Balance Beam, and Overcooked [5, 34, 26]. These three environments will help illustrate the differences between statistical diversity, cross-play diversity, and mixed-play diversity. In particular, the Blind Bandits highlight the importance of reward-alignment that is present in cross-play but not statistical diversity. The Balance Beam environment demonstrates the pitfall of handshakes during cross-play minimization, and how this is mitigated with mixed-play. Finally, the Overcooked environments demonstrate the scalability of CoMeDi, and allow us to evaluate the generated set of diverse agents by training convention-aware policies to play with real human partners. More detailed descriptions of all experiments are in the appendix.

### 4.1 Blind Bandits

In the Blind Bandits environment, we extend the classic multi-armed bandit problem into a two-player game. The main twist is that each agent cannot observe the actions of their partner during the game, so they cannot change their strategy in reaction to their partner. It is a collaborative 2-player game

| $\beta$ | SP $\uparrow$ | XP $\downarrow$ | HS $\downarrow$ | PX $\downarrow$ | LS $\uparrow$ | RS $\downarrow$ |
|------|-------|--------|-------|-------|-------|--------|
| 0.00 | 1.616 | **-0.392** | 0.16 | **0.00** | 1.128 | **-0.656** |
| 0.25 | 1.808 | 0.096 | **0.00** | **0.00** | 1.272 | 0.096 |
| 0.50 | **2.000** | 0.200 | **0.00** | **0.00** | **1.384** | 0.128 |
| 1.00 | 1.904 | 0.632 | **0.00** | 0.24 | 1.232 | 0.160 |

Table 1: Results of Mixed-Play in the Balance Beam Environment: SP is the self-play score. XP is the cross-play score with the first convention. HS (handshake) is the fraction of cross-play games where a player moves off the line, while PX (perfect XP) is the fraction of with a perfect 2.0 score. LS and RS are the scores under cross-play with the left-biased and right-biased hand-coded agents.

where each player takes $k$ steps, $k \geq 2$, and at each step they have to choose to go left (L) or right (R). The possible trajectories when $k = 2$ are represented in Fig. 4.

An important characteristic of the Blind Bandits environment is that all conventions that converge to $S$ are compatible with one another while being incompatible with the only convention that converges to $G$. Therefore, if we want a diverse set of 2 conventions, we would like one convention to converge to $S$ and another to converge to $G$.

**Baseline Results for Blind Bandits.** Existing statistical divergence techniques, like ADAP, typically converge to $S$ conventions because it affords a larger space of policies. Even when trying to train more conventions, they could all converge to $S$ conventions while still satisfying statistical diversity, since there are multiple possible distinct policies associated with $S$. Unfortunately, this gives a set of agents that are diverse with respect to trajectory metrics, but not with respect to the task, since the statistical diversity policies cannot lead to an agent compatible with the $G$ convention.

**CoMeDi Results for Blind Bandits.** By minimizing cross-play, we can reliably train a set of agents that converge to both the $S$ and $G$ conventions. First, we train a single agent via self-play to converge to one of the $S$ conventions. Then, under the cross-play regularization (with $\alpha = 1.0$), the reward payoff for a second agent is a transformed environment where each self-play reward is subtracted by the cross-play reward with the first agent. All policies with a self-play score of $S$ would have a score of 0 in the transformed environment, but all policies with a self-play score of $G$ would still have a score of $G$ in the transformed environment, so an agent trained on this transformed environment would converge to $G$. Indeed, as we see in right plot of Fig. 5, the second convention consistently converges to $G$.

### 4.2 Balance Beam

The Balance Beam environment is a collaborative environment with two players on a line that has 5 locations with unit spacing, illustrated in Fig. 6. Both agents move simultaneously and they get a high reward if they land on the same space after their actions, but they must move from their current location at each timestep. As a human-like test set of policies, we hand-design a left-biased agent and a right-biased agent. Fig. 6 shows a visualization of the environment.

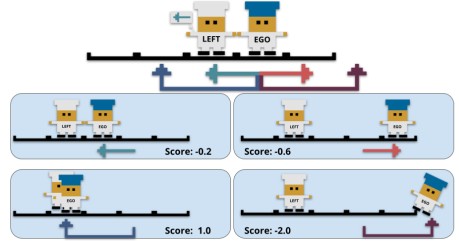

Figure 6: Example of the Balance Beam environment. The agent with a white hat follows the left-biased policy. The blue agent has four different actions it can choose. If an agent steps off the line, the game ends with a large negative score.

As opposed to Blind Bandits, the Balance Beam environment involves multiple timesteps and is prone to handshake formation. As we will show in the next section, mixed-play allows us to combat the handshake formation of cross-play diversity in the Balance Beam environment.

**Balance Beam Results.** We train the first convention with base MAPPO, which converges to a self-play score of 2.0. The expected cross-play of this first convention with the hand-coded left-biased

agent is 0.384 while the score is 1.368 with the right-biased agent, so it acts very similar to the right-biased agent.

Ideally, we would want a second convention to have a high cross-play score with the left-biased agent and a low cross-play score with the right-biased agent so it would ensure that human players from either convention could find a compatible AI partner. Its self-play score should be 2.0, indicating an optimal equilibrium convention. Also, it should never take actions that are clearly incorrect, like stepping off of the line, which would result in a large negative score along with the early termination of the episode.

Although pure cross-play minimization ($\beta = 0$) results in the lowest cross-play scores (as shown in Table 1), we notice signs of handshakes: 16% of the cross-play games result in the action of stepping off of the line, as indicated in the HS column. Moreover, pure cross-play minimization does not give perfect self-play scores, indicating that it takes suboptimal actions in some states to establish a handshake. When the mixed-play weight is set to a low value ($\beta = 0.25$), the agent no longer steps off the line when the convention is breached, but it still results in an imperfect convention since it takes suboptimal actions to minimize cross-play. If the mixed-play weight is set to a high amount ($\beta = 1.0$), the agent follows an identical trajectory to the first convention in 24% of the games, indicating that it is overcorrecting the issue of handshakes. The best convention we found had $\beta = 0.5$ as shown in Table 1, which was able to generate a perfect convention that does not exhibit signs of handshakes. This convention also has the highest cross-play with the left-biased agent, indicating that mixed-play enables us to generate an agent that is more aligned to human notions of "natural" conventions since this agent always acts in good faith instead of conducting handshakes.

From the results above, we empirically demonstrate that mixed-play addresses the issue of handshakes, generating a pair of meaningfully diverse agents.

### 4.3 Overcooked

Overcooked is a fully cooperative game where two chefs controlled by separate players try to cook and deliver as many dishes of food as possible in a given amount of time [5, 26]. The main challenge is to coordinate by subdividing tasks and not interfering with each other's progress.

For our experiments, we generate 2 baseline agents and 2 convention-aware agents using CoMeDi. The baselines are "SP" and "ADAP" which refer to the base MAPPO algorithm and training a convention-aware agent on ADAP. Our agents are "XP" and "CoMeDi" which refer to the convention-aware agents on CoMeDi where $\beta = 0.0$ (pure cross-play minimization) and $\beta = 1.0$ respectively.

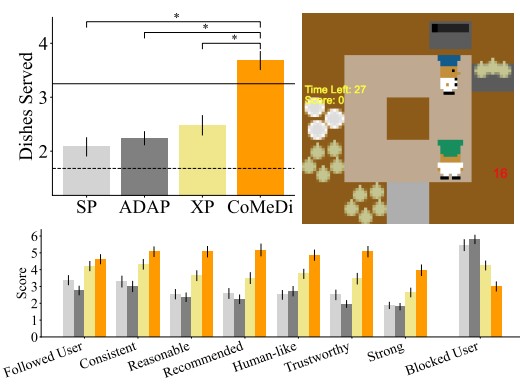

Figure 7: Results for the Coordination Ring: average scores (top left), environment visualization (top right), and user survey feedback using the 7-point Likert scale (bottom). Higher average scores are better. The dashed line is the average score single-player human score while the solid line is the average score when paired with an expert human. Higher is better for all but the last feedback score. Error bars represent the standard error.

**Overcooked User Study.** In both the Blind Bandits and Balance Beam environments, we could construct an artificial test set by designing human-like conventions. However, this is infeasible for Overcooked because humans can exhibit unpredictable dynamics, such as adapting to partners. Therefore, we test the quality of our four agents by experimenting on a population of 25 human users. The results of the user study in the Coordination Ring environment are in Fig. 7. In this environment, players need to coordinate on a strategy so they do not bump into one another. A full description of the environment, simulation results, and additional user-study results in the Cramped Room layout are in the appendix. Videos of users interacting with each agent can be found on our website.

| 1 | 2 | 3 | 4 | 5 | 6 | 7 | 8 |
|------|------|------|------|------|------|------|------|
| 0.50 | 0.00 | 0.25 | 0.00 | 0.00 | 0.05 | 0.04 | 0.15 |

Table 2: Frequency of each convention being the most likely in the encountered "convention-dependent" observations in Coordination Ring.

In terms of scores, we see that CoMeDi (score of 3.68) performs best, followed by XP (2.48), ADAP (2.24), and SP (2.08). The average score when playing with an expert human partner is 3.25, so only CoMeDi surpasses human-level performance ($p = 0.045$). In terms of statistical significance, CoMeDi outperforms all baselines with $p < 0.001$. CoMeDi also was strongest across all metrics in the survey. Users wrote that CoMeDi "knew exactly what to do next," with one user even claiming that it "is GODLIKE" and "adapted to my moves like he can see into the future." Sentiments were mixed for XP with some users reporting that it was "doing great decisions, but sometimes he seemed confused." With ADAP and SP, users complained that the bots would "block" their progress often and "just stood around the onions and did nothing."

**Diversity of the Human Test Set.**   Although users tended to favor CoMeDi across the metrics we measured, we wanted to determine why CoMeDi performed well. Specifically, we wanted to determine whether humans actually followed a diverse set of conventions and whether CoMeDi's strength comes from finding a specific human-like convention or covering a larger observation space.

To answer these questions, we conducted an experiment where we explicitly predict the probability of being in a CoMeDi convention conditioned on the observation. Since our usage of QDagger can be approximated as a gated mixture-of-experts model, we can explicitly train a "gate" network as a classifier that determines which convention an observation belongs to. Out of the 5000 observations that CoMeDi encountered when working with users in Coordination Ring, 2349 of them were aligned with a specific convention with probability greater than 50%, which we refer to as "convention-specific" observations. The frequency of most likely conventions is reported in Table 2.

Although convention 1, which is equivalent to the pure self-play policy, behaves well in 50% of the convention-specific observations, knowledge of other conventions is still vital. Furthermore, not all players aligned with convention 1. For instance, one player had 65% of their convention-specific observations align with convention 8 while only 27% aligned with convention 1. When calculating which convention each player followed a plurality of the time, 20 players followed convention 1, three followed convention 8, and two followed convention 3. No user perfectly followed a single convention, indicating that understanding multiple conventions made CoMeDi more robust overall.

These results demonstrate that there is non-trivial diversity in the human distribution of conventions and that CoMeDi's strengths come from discovering multiple human-like conventions along with having a more robust policy to handle deviations from these conventions.

## 5   Conclusion

In this work, we introduce CoMeDi, an algorithm for generating diverse agents that combines the idea of minimizing cross-play with existing agents to generate novel conventions, and the technique of mixed-play to mitigate optimization challenges with cross-play. Compared to prior work that defines the difference between conventions as a difference in generated trajectory distributions, our work uses cross-play as a measure of incompatibility between conventions. As seen in the Blind Bandits environment, cross-play gives more information than statistical diversity techniques since it can identify whether conventions are redundant. In the Balance Beam environment, we show how mixed-play addresses the issue of handshakes by incentivizing agents to act in good faith at each state in the environment. Finally, we analyze Overcooked and see how a convention-aware agent trained with CoMeDi outperforms prior techniques with users, achieving expert human-level performance.

Although we use a simple BC-based algorithm for training a convention-aware agent, future work could utilize the diverse set of conventions generated by CoMeDi more effectively to create an agent that uses the history of past interactions to dynamically adapt to a user's convention.

## 6 Acknowledgements

This research was supported in part by AFOSR YIP, DARPA YIP, NSF Awards #2006388 and #2125511, and JP Morgan. We would also like to thank Jakob N. Foerster for his useful discussion with us regarding handshakes and Hengyuan Hu for clarifying the differences between CoMeDi and ADVERSITY.

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

# A  CoMeDi Algorithm Details

## A.1  Mixed-play Buffer Collection

Mixed-play consists of two phases in each episode: mixed-state generation and self-play. The "input" policies are the policy for the convention we are currently training $\pi_n$ and the partner policy used for cross-play optimization $\pi^*$ (using the same notation from Eq 7). First, we choose a random timestep within the episode that represents the length of the first phase (Line 2). Until this timestep occurs, we randomly sample the action from self-play or cross-play for both players (Lines 5-9). We do not store any of these transitions in the training buffer. Instead, we use the state at the last timestep and pretend that this is the initial state of the environment. For the rest of the timesteps, we perform the second phase, by taking self-play actions and store that in the buffer (Lines 10-12). When optimizing, we treat this new buffer the same as we would treat self-play, but modified with a positive weight hyperparameter, $\beta$, representing the importance of mixed-play.

---

**Algorithm 1:** Generating Mixed Play Buffer

**Input:** policies $\pi_n, \pi^*$, MDP $M$
**Output:** Replay buffer from running mixed-play.

1   $s_0, R \leftarrow s, 0$          // start state, reward
2   $t \sim \text{uniform}(1, T)$       // length of mixed-state generation phase
3   **for** $i \leftarrow [0, T)$ **do**
4     $o^1, o^2 \leftarrow o(s_i)$
5     **if** $i < t$ **then**
6       $\pi_1^m \leftarrow$ Randomly choose $\pi_n$ or $\pi^*$
7       $\pi_2^m \leftarrow$ Randomly choose $\pi_n$ or $\pi^*$
8       $a_i^1, a_i^2 \leftarrow \pi_1^m(o^1), \pi_2^m(o^2)$
9       $s_{i+1}, - \leftarrow$ Step in $M$ with$(a_i^1, a_i^2)$
10    **else**
11      $a_i^1, a_i^2 \leftarrow \pi_n(o^1), \pi_n(o^2)$         // self-play
12      $s_{i+1}, r_i \leftarrow$ Step in $M$ with$(a_i^1, a_i^2)$    // keep reward in phase2

**Return:** ReplayBuffer$(s_{t:T}, a_{t:T}^1, a_{t:T}^2, r_{t:T})$

---

## A.2  Full CoMeDi Algorithm

Simplified pseudocode for the CoMeDi algorithm is presented below. Note that conventions are generated in a sequential order, with $\pi_1$ being trained with standard self-play and each $\pi_i$ being trained with the awareness of prior conventions, $D_{1:i-1}$. The arg max operation in line 4 is estimated empirically by simulating a fixed number of rounds of cross-play in the environment with each existing convention and selecting the convention with the highest cross-play as $\pi^*$.

---

**Algorithm 2:** Diverse Conventions with CoMeDi

**Input:** Number of policies to generate $n$
**Output:** Diverse set of conventions $D$

1   $D \leftarrow (\pi_1, \ldots, \pi_n)$, parameterized by $(\theta_1, \ldots, \theta_n)$
2   Train $\pi_1$ with standard self-play
3   **for** $i \in \{2, \ldots, n\}$ **do**
4     **while** policy $\pi_i$ has not converged **do**
5       $\pi^* \leftarrow \arg\max_{\pi^* \in D_{1:i-1}} J(\pi_i, \pi^*)$
6       $\tau_{SP} \leftarrow \texttt{GetRollout}(\pi_i, \pi_i)$
7       $\tau_{XP} \leftarrow \texttt{GetRollout}(\pi_i, \pi^*)$
8       $\tau_{MP} \leftarrow \texttt{MixedPlayRollout}(\pi_i, \pi^*)$
9       Estimate $J(\pi_i, \pi_i), J(\pi_i, \pi^*), J_M(\pi_i, \pi^*)$ with $\tau_{SP}, \tau_{XP}, \tau_{MP}$
10      $\theta_i \leftarrow \theta_i + \nabla_{\theta_i}[-J(\pi_i, \pi_i) + \alpha J(\pi_i, \pi^*) - \beta J_M(\pi_i, \pi^*)]$

**Return:** $D$

---

## A.3 Implementation Details

We base the implementation of our algorithm on the Multi-Agent PPO algorithm (MAPPO) [36]. MAPPO is an actor-critic method which, in standard self-play, trains a single actor network for the policy and a single critic network for the value function [25]. To adapt MAPPO to train a pool of $n$ conventions using our proposed mixed-play algorithm, we train $n$ actor networks, $n$ self-play critic networks, and $n^2 - n$ cross-play critic networks, each representing a cross-play pairing between the $n$ conventions. We also use the PantheonRL library [24] to design our environments and training algorithms since it is designed to handle dynamic training interactions like cross-play and mixed-play. We have also integrated CoMeDi with a new GPU-accelerated simulation framework [26], which enables the collection of large batches of cross-play and mixed-play buffers in parallel with the collection of self-play buffers.

Moreover, instead of training the whole batch of $n$ diverse agents in parallel, in practice we sequentially grow the set of agents one at a time, keeping the previous agents fixed. We find that sequential generation leads to more stable training: since the previous agents are fixed, the diversity regularization term becomes a reward shaping term that is only a function of the policy of the current agent.

## A.4 Practical Guidelines for Hyperparameter Tuning

There are some safe choices for hyperparameters that work well in general, which we used to tune the hyperparameters for our experiments. First, we observe that directly using the best MAPPO hyperparameters for the particular environment, like learning rate and the model architecture, transfers well to CoMeDi. To find the cross-play weight ($\alpha$), fix the mixed-play weight to 0 and find the lowest value for the cross-play weight such that increasing it further does not significantly increase the self-play score or decrease the cross-play score. If the cross-play weight is too high, this may cause training instabilities since the updates to increase the self-play score would be directly counteracted by the updates to decrease the cross-play score. Finally, choose the value of the mixed-play parameter such that the average mixed-play score (for the second half) is slightly less than half of the self-play score, which would indicate that self-play is able to smoothly continue from any mixed-play state.

The guidelines for choosing hyperparameters works well in general, but domain knowledge of the environments also helps. If your specific environment also has some indicators of handshakes, you can also use those to determine if handshakes are still happening. Furthermore, environments where partners' actions are not visible, like Blind Bandits, do not require mixed-play at all because handshakes are impossible. In practice, we have seen that CoMeDi is relatively robust to hyperparameters and it gives reasonable policies with a cross-play weight of 0.5 and a mixed-play weight of 1, even if they are not perfectly optimal.

Choosing a population size is also an art, but due to the sequential nature of CoMeDi, prior conventions are unaffected by the generation of later conventions. The choice of algorithm for generating a "convention-aware agent" would likely influence the number of conventions to use for the diverse set.

## A.5 Extending to Larger Teams

When using CoMeDi for cooperative games with more than 2 players, we can follow the same algorithm presented in 2, but we have to be a bit careful when collecting rollouts.

To collect the cross-play buffer between $\pi_i$ and an existing $\pi^*$ in a $k$-player game, we can randomly assign each player to one of the conventions but keep those assignments consistent throughout the duration of the episode. The same logic regarding the minimization of cross-play rewards still applies since semantically similar conventions would result in a high reward even when the team contains a mix of the conventions.

To collect the mixed-play buffer, we would randomly choose between the two conventions for each player at each timestep during the mixed-state generation phase. We would still treat self-play as normal by using the convention being trained as the convention for all players.

# B Experiments

## B.1 Choice of Baselines

Throughout this work, we compare the performance of CoMeDi against pure MAPPO, a modified version of ADAP, and a pure cross-play minimization baseline (CoMeDi with $\beta = 0$). We do not directly use ADAP because we find that its generative capabilities from parameter sharing often limits the diversity of its agents, resulting in very high cross-play between agents in the population. By adding its diversity loss to the MAPPO algorithm, we eliminate the confounding variables of the base PPO implementation and the parameter sharing in the original algorithm. We also do not directly compare against TrajeDi because a fundamental aspect of its algorithm is the concurrent generation of a common best response agent, which implicitly optimizes for high cross-play within the "diverse set" of policies. However, upon analyzing TrajeDi's diversity loss, we note that it is very similar in practice to ADAP's loss. Therefore, we believe that our modified version of ADAP is the fairest representative of statistical diversity approaches. Concurrent work that implements a technique similar to our pure cross-play minimization provides some other benchmarks with MAVEN and TrajeDi which show similar results to what we experienced with our modification of ADAP.

We do not compare our work to those in section 2.3, because they require more assumptions regarding the training domain. In particular, the strength of reward shaping methods (and other domain engineering techniques) is highly dependent on the manual design of the appropriate environment parameter space. However, CoMeDi can be interpreted as an *automatic* technique for reward shaping to form diverse conventions, which can potentially eliminate the need to engineer the environment parameter space for domain randomization.

## B.2 Hyperparameters

Table 3: Common hyperparameters for agents in Blind Bandits and Balance Beam

| hyperparameters | value |
|---|---|
| fc layer dim | 512 |
| num fc | 2 |
| activation | ReLU |
| network | mlp |
| ppo epochs | 15 |
| mini batch | 1 |

Table 4: Hyperparameters in Blind Bandits

| hyperparameters | value |
|---|---|
| buffer length | 200 |
| environment timesteps | 10000 |
| actor/critic lr | $2 \times 10^{-5}$ |
| linear lr decay | False |
| entropy coef | 0.01 |
| ADAP $\alpha$ | 0.2 |
| CoMeDi $\alpha$ | 1.0 |
| CoMeDi $\beta$ | 0.0 |

Table 5: Hyperparameters in Balance Beam

| hyperparameters | value |
| --- | --- |
| buffer length | 1250 |
| environment timesteps | 50000 |
| actor/critic lr | $2.5 \times 10^{-5}$ |
| linear lr decay | True |
| entropy coef | 0.01 |
| ADAP $\alpha$ | 0.05 |
| CoMeDi $\alpha$ | 0.3 |
| CoMeDi $\beta$ (ablation) | 0.0, 0.25, 0.5, 1.0 |

Table 6: Hyperparameters in Overcooked (only training set)

| hyperparameters | value |
| --- | --- |
| CNN Kernel Size | $3 \times 3$ |
| fc layer dim | 64 |
| num fc | 2 |
| activation | ReLU |
| rollout threads | 50 |
| buffer length (per thread) | 200 |
| environment timesteps | 1000000 |
| ppo epochs | 10 |
| actor/critic lr | $1 \times 10^{-2}$ |
| linear lr decay | True |
| entropy coef | 0.0 |
| ADAP $\alpha$ | 0.025 |
| CoMeDi $\alpha$ | 0.5 |
| CoMeDi $\beta$ | 0.0, 1.0 |

Table 7: Hyperparameters in Overcooked (convention-aware agents)

| hyperparameters | value |
| --- | --- |
| CNN Kernel Size | $3 \times 3$ |
| fc layer dim | 64 |
| num fc | 2 |
| activation | ReLU |
| rollout threads | 50 |
| buffer length (per thread) | 200 |
| environment timesteps (SP phase) | 200000 |
| ppo epochs (SP phase) | 100 |
| lr | $1 \times 10^{-2}$ |
| entropy coef | $1 \times 10^{-3}$ |

### B.3 Compute Resources

We conducted our experiments with our lab's internal cluster. We only used the Intel Xeon Silver 4214R CPU for training in Blind Bandits and Balance Beam. For the Overcooked experiments, we used an additional NVIDIA TITAN RTX, which required around 3 hours per configuration. However, this time can be reduced significantly by disabling deterministic behavior in CUDA. Current work on optimizing performance with the GPU-accelerated simulator also shows that this time can be reduced.

### B.4 Blind Bandits Environment Description

The Blind Bandits environment is a collaborative two-player MDP where each player takes $k$ steps, $k \geq 2$, and at each step they have to choose to go left (L) or right (R). Each player is given their own past history of actions in the episode, but cannot see the others' actions. There are two ways to get a positive score. To get a score of $S$, the first player's first step must be L, and the second player's last step must be L. To get a score of $G > S$, the first player's first step must be R, but all following steps by all players must be L except for the second player's last step, which must be R. If the players fail to coordinate, they get a score of 0.

The policies that converge to $S$ are all compatible with one another since all of them have agent 1 start with L and have agent 2 end with L (shown in blue). If two agents are playing randomly, there is a 0.25 probability that they will get a reward of $S$. Meanwhile, only one trajectory will result in a score of $G$ (shown in orange), so there is only a $2^{-2k}$ chance that random agents will get a reward of $G$. Notice how these two conventions ($S$ and $G$) are entirely incompatible with one another because their first and last moves must be different, and are therefore different equilibria. Ideally, we want to find a representative policy for both conventions using a technique that finds a diverse set of conventions.

### B.5 Blind Bandits Baseline Results

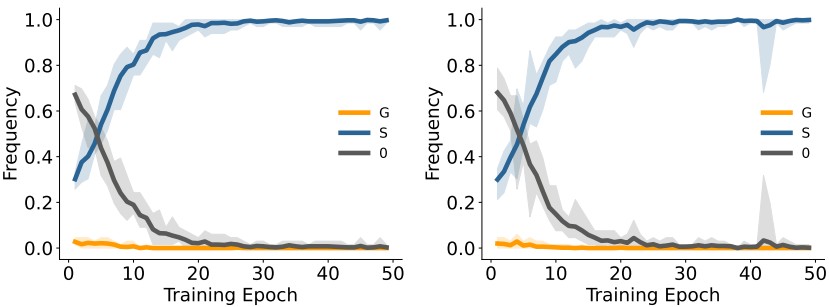

Figure 8: Frequency of self-play scores during the optimization of two conventions using statistical diversity (the ADAP loss) from 10 independent seeds. The blue line indicates an $S$ convention, the orange line indicates an $G$ convention, and the black line indicates a score of 0 (all other paths that lead to 0). With ADAP, neither convention converges to $G$. We choose $k = 3$ as the number of steps in the environment.

In Figure 8, we see that using ADAP to train a set of two agents almost always leads to them both converging to $S$. For this result, we used the ADAP diversity weight of 0.2, the most common value used in the original paper. Increasing this weight sometimes results in discovering a few $G$ conventions, but this is inconsistent and results in more unstable training.

In fact, training an agent to converge to the $G$ convention is more difficult that it appears. Off-the-shelf MARL algorithms typically converge to the $S$ convention as well. At the first training epoch, the actor chooses random actions, so it will get a score of $S$ 1/4 of the time while it gets a score of $G$ with a probability of $2^{-2k}$. The critic network (in both the decentralized and centralized setting) will learn that the value of choosing to go left in the first state is $S/2$ while the value of choosing to go to the right is $G/2^k$. Therefore, the policy updates will favor the $S$ convention in the first epoch as long as $S > G/2^{k-1}$, and this separation gets larger until it fully converges to an $S$ convention.

Most existing zero-shot coordination (ZSC) algorithms would also converge to the $S$ convention. The TrajeDi algorithm finds a common best response to a set of conventions that have a wide diversity in trajectory, but the $G$ convention has only one trajectory so it will never be chosen. The Off Belief Learning (OBL) algorithm would also only converge to the $S$ convention because it acts on the belief that its partner is a random agent. Higher levels of OBL will be stuck in the $S$ convention since they assume that their partner is from a lower level of OBL. Note that the purpose of ZSC is to have a high cross-play between independent runs of the same algorithm, so both of these techniques succeed in that manner, but this does not imply that these algorithms would have high cross-play with humans.

## B.6 Balance Beam Environment Description

At the start of the game, the location of each player is randomly initialized. At each timestep, both players take simultaneous actions, and can move 1 or 2 locations to the left or right, but cannot stay still. If an action leads to them falling off the line, they get a score of -1 multiplied by the number of remaining timesteps. Otherwise, they get a score of $-d(s_1, s_2)/5$ where $d$ is the distance between the two player's locations. Finally, if both players are on the same spot at the end of their turn, they get an extra point. Each episode lasts two timesteps, and the players see the result of the first step's actions before making their second move.

A perfect convention would always get a score of 2.0, because players can score +1 in each of the two timesteps. The worst score is -2.0, which occurs when one player moves off of the line at the first time step.

We also hand-code some conventions to see if CoMeDi discovers conventions that are similar to those that humans follow. There are two simple conventions: a left-biased convention and a right-biased convention. These dictate how ties are broken when multiple actions are equally optimal. For example, if the distance between two players is 1 step, like in Figure 6 of the main text, the player from a left-biased convention would want to move to the open space to the left since staying in one spot is not an option.

## B.7 Balance Beam Baseline Results

We designed the Balance Beam environment to distill the issue of handshakes when minimizing cross-play, which is why the main text only emphasizes the impact of the $\beta$ hyperparameter in CoMeDi. However, we also trained ADAP in this environment to see how it compares to the other approaches.

The first trained convention gets a score of 2.0, and gets an expected score of 0.768 and 1.112 when paired with the left and right-biased agents respectively.

The second trained convention gets a score of 1.808, and gets an expected score of 1.048 and 0.176 when paired with the left and right-biased agents respectively. Its cross play score with the first trained convention is 0.392.

When training the ADAP agents, we observed a very unstable training process resulting in very low self-play scores with typical values of the diversity weight parameter. For this reason, we had to choose a relatively low value for the loss parameter (0.05) in order to make a fair comparison with our technique.

## B.8 Overcooked Agent Generation

For our experiments, we generated 2 baseline agents and 2 convention-aware agents using CoMeDi. The first baseline agent, which we refer to as "SP", was trained with pure self-play, and we tuned the hyperparameters to maximize its self-play score. For all other generated agents, we maintained the same hyperparameters that are inherent to MAPPO, but we would tune the diversity weights specific to each algorithm. Our second baseline, which we refer to as "ADAP", is a convention-aware agent to a population of 8 agents trained with ADAP with no parameter sharing and a diversity weight of 0.025. The XP agent was also a convention-aware agent to a population of 8 agents trained with $\alpha = 0.5$ and $\beta = 0.0$. The CoMeDi agent was the same as XP, but its population was trained with $\alpha = 0.5$ and $\beta = 1.0$.

For each layout of Overcooked, we can determine the expected number of dishes to be served by each agent in self-play. In the Cramped Room, SP averages 4.36 dishes, ADAP averages 2.75 dishes, XP averages 4.68 dishes, and CoMeDi averages 5.52 dishes. Meanwhile, in the Coordination Ring, SP averages 3.47 dishes, ADAP averages 1.90 dishes, XP averages 3.06 dishes, and CoMeDi averages 5.36 dishes.

We can also calculate the expected reward for the best and worst performing agents in the training sets of ADAP, XP, and MP. In Cramped Room, ADAP's average rewards spanned 0 to 5.98, XP's rewards spanned 4.12 to 5.96, while MP's rewards spanned 5.0 to 5.88. In Coordination Ring, ADAP's average rewards spanned 0 to 4.965, XP's rewards spanned 2.75 to 4.56, while MP's rewards spanned 4.92 to 5.99.

When training agents with ADAP, we would frequently see a few policies with very high expected returns with the remaining policies having low scores. We attempted to tune ADAP's diversity weight to enable more balanced generation, but this issue continued to persist even with a final diversity weight significantly lower than the values presented in the original paper.

## B.9  Overcooked User Study Setup

The human-AI interaction portion of this research was approved by our IRB.

Our total population consisted of 25 paid participants with varying prior experiences in Overcooked, recruited through Prolific. We did not impose any conditions on participation through Prolific except for requiring "proficiency in English language." We paid $10.33 (US dollars) per participant for approximately 40 minutes of time, equivalent to $15.50 per hour. We titled the study "Playing with AI Agents in the Overcooked Video Game" with the following description:

"In this study, participants will play with 4 different AI agents in 2 settings of the Overcooked game. Our goal is to understand how well trained agents can work with humans in tasks that require coordination. Your task is to try to work with the AI to cook many "dishes of onion soup" within a 40-second time limit. You will also fill out short surveys to judge the quality of the AI agents. You will play 20 games in total.

To play the game, you need to use a keyboard with arrow keys and a space bar. Desktops, laptops, and tablets with keyboards may be used. The AI agents runs on your browser and are designed to be lightweight so they should be compatible with most hardware.

The games will be fast-paced, and we may reject submissions that are consistently unable to score points in the game. As long as we can see that you are trying to score points, your submission will be accepted.

NOTE: Please only accept this study if you have at least 8 GB RAM. The game will take up to 2 GB of RAM since we are loading models onto your computer so you have a smoother experience. If the game freezes, please let us know and refresh the page. Please use Firefox, Chrome, or Safari."

To ensure that participants across the world are able to complete the study without excessive latency, we run all models on the users' devices directly through tensorflow-js.

Each user played 2 games in a layout independently before playing each AI agent for two 40-second rounds in a random order. The users first played with all agents in the Cramped Room environment before playing with all agents in the Coordination Ring environment.

We also asked the users to fill out a short survey with qualitative questions about each partner after playing both rounds in any configuration using the 7-point Likert scale.

We presented the following 8 statements (in order):

1. The AI followed my lead when making decisions.
2. The AI agent frequently blocked my progress.
3. The AI was consistent in its actions.
4. The AI always made reasonable actions throughout the game.
5. I would like to collaborate with this AI in future Overcooked tasks.
6. The AI's actions were human-like.
7. I trusted the AI agent in making good decisions.
8. The AI agent was better than me at this game.

Note that a higher Likert score is better for all of the prompts except for the second question.

We also presented an optional free-response section with "Other comments or observations?" that users could fill out at the end of each survey.

To determine statistical significance, we use the paired Student t-test between the scores across different AI agents for each layout.

We also asked a smaller set of users (8 users) to play with an expert human player in-person after playing with all AI agents to determine what human-level performance would entail without allowing

explicit communication. This expert human player was trained to adapt to different play styles, and users reported that this human player was "very good" at the game, noting fast reflexes. This set of users was mutually exclusive from the Prolific set, since the games needed to be played in person. We determine if there is a statistically significant difference between the AI agents and the human expert through the unpaired t-test.

## B.10 Cramped Room Overcooked Results

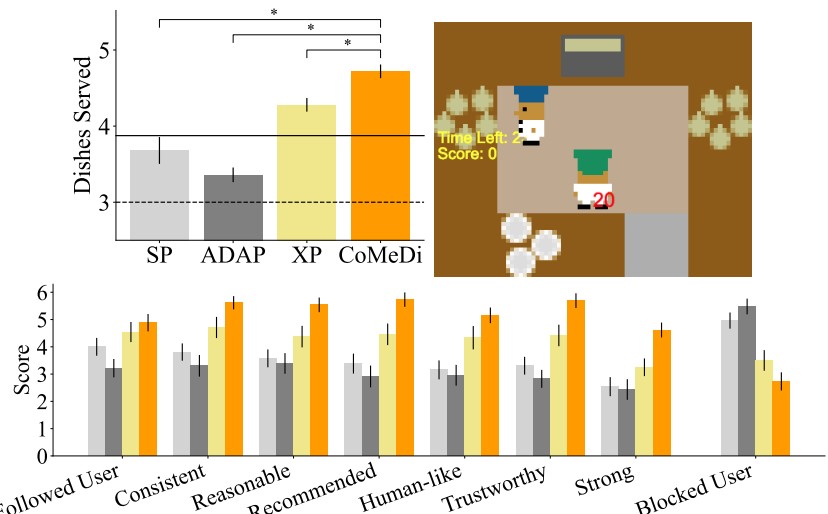

Figure 9: Results for the Cramped Room: average scores (top left), environment visualization (top right), and user survey feedback using the 7-point Likert scale(bottom). Higher average scores are better. The dashed horizontal line indicates the average score when a player is working alone while the solid horizontal line indicates the average score when paired with an expert human. Higher is better for all but the last feedback score. Error bars represent the standard error.

In the Cramped Room Environment, players have a very small area to move around in, and they need to coordinate around the management of ingredients and the usage of plates. Given the simplicity of the setting (relative to the Coordination ring), humans might be able to adapt quickly to AI players since the patterns of movement can be very clear after interacting for a short time. In particular, we note that this layout has lower convention dependence than the Coordination Ring.

The results of the user study in the Cramped Room environment are in Figure 9. In terms of scores, we see that CoMeDi (score of 4.72) performs the best, followed by XP (4.28), SP (3.68), and ADAP (3.36). In terms of statistical significance, CoMeDi outperforms all AI agents ($p < 10^{-3}$) and the expert human ($p < 10^{-4}$). XP also outperforms ADAP, SP, and the expert human ($p < 0.01$ for each).

We also analyzed the convention-specific observations in the Cramped Room environment. Out of 5000 observations, 2124 observations were aligned to a specific convention with probability over 50%. We show the frequency of each convention being the most likely one below:

| 1 | 2 | 3 | 4 | 5 | 6 | 7 | 8 |
|------|------|------|------|------|------|------|------|
| 0.05 | 0.25 | 0.00 | 0.01 | 0.45 | 0.11 | 0.04 | 0.09 |

Table 8: Frequency of each convention being the most likely in the encountered "convention-dependent" observations in Cramped Room.

When considering which convention each player followed a plurality of the time, 16 players followed convention 5, 8 players followed convention 2, and 1 player followed convention 6.

## B.11 User Free Response Feedback

User feedback for SP in the Cramped Room environment:

- This AI trusted me more than the others. He waited for me to get the plate and deliver the soup. The other bots rushed to do it themselves.
- I could not understand what the AI was doing
- The AI was slow in decision making
- Ai was a bit hesitant sometimes

User feedback for ADAP in the Cramped Room environment:

- The AI sometimes blocked me. He held the onion in his hands, and it was already cooked, and I had the plate in my hands, but I couldn't get to pick up the soup because he was blocking it. I think he was trying to place the onion even tho there was no space to place it in.
- AI was blocking me whole time
- i didnt like this one...
- It was somehow worse than S, it blocked the plate for like 7 seconds
- the ai seemed very confused
- He tried to do things right, but sometimes he seemed confused.

User feedback for XP in the Cramped Room environment:

- This AI was more lazy. He only held the plate in his hand and waited for me to cook the soup. The other AI wasn't that way (AI D). He cooked and adapted in getting the plate and everything. This bot did the exact opposite.
- I did not like the fact that it could not path find a different way to deliver the full plate.
- The AI seems to be faster
- ai was very good at this game
- Once he picked up a plate after I got one plate, then he never stopped that action until I put the next 3 onions in the pot.

User feedback for CoMeDi in the Cramped Room environment:

- This bot was by far the best. He was very consistent in everything he did, his moves were all correct and I felt very good, working with him.
- this one was not so bad
- This AI could work by himself, literally, and get the same 100 score, I believe.

User feedback for SP in the Coordination Ring environment:

- The AI did the correct thing, but when it came to pressing the space bar, he struggled. Even tho he just needed to press the space bar, and he stood right in front of it, and it was just one space bar away, he decided to wait in front of it for some seconds and then press it, which concluded us to make less soup overall.
- the first soup was ok and then AI like "freeze"...it was weird.
- ok so this one completly stood in my way while i was trying to get the plate he just stood there with an onion on his hand. Aside from the ai i really likes this study and experiment and hope one day ill be able to participate in one of them again!
- At first it looked like the cooperation was working until when it just stopped doing anytihng and occasionally just blocked the way to the stoves. Towards the end it just standed in the corner doing nothing.
- Yes it was consistenly bad

- He didn't know what to do when picking a plate and there were onions missing in a pot.

User feedback for ADAP in the Coordination Ring environment:

- This bot blocked me a lot of times. He also just stood around the onions and did nothing, which stopped us both. He sometimes got a plate in his hand and just went up and down all the time until he decided to play normally again. Would not play with this bot.
- i also didnt like this ai.
- This one was very bad at making decisions and pathfinding
- the AI slowed the process
- AI was better than the other ones so far, but still got alot in the way which is frustrating to play with.
- He was in the way a lot and almost never knew what to do next, or where to go.

User feedback for XP in the Coordination Ring environment:

- He did very good in the first round, we always went clockwise. It was perfect. In the second round, he decided to be the lazy chef. He was too lazy to pick up an onion or the plate.
- It's moves felt very scattered and didn't make sense to me.
- The AI kept getting in the way and got nothing done.
- The AI was helpful
- This one was doing great decisions, but sometimes he seemed confused. In general I liked it.

User feedback for CoMeDi in the Coordination Ring environment:

- This bot is GODLIKE! He did everything correct, he adapted to my moves like he can see into the future. Just great playing with him. The most efficient by far.
- it seems sometimes you get into a flow, which the AI breaks after a while
- He knew exactly what to do next.
- The experiment was very cool because every AI was unique. The best AI was AI M by far, my most favorite. If I was a chef, I would definetly hire him!

### B.12   Additional Overcooked Simulation Experiments

Our Human-AI experiments only covered the Coordination Ring (random1) and Cramped Room (simple) environments, since these are considered the canonical Overcooked layouts. However, we also trained models for 3 other layouts used in the original Overcooked-AI papery: Counter Circuit (random3), Asymmetric Advantages (unident_s), and Forced Coordination (random0). To train the new models, we use the same hyperparameters as the other Overcooked models. We used the tensorflow-js versions of the PPO_BC and PBT models as held-out test conventions.

When paired with the PBT partner, we get the following average scores:

| layout | CoMeDi | XP | ADAP | SP | PPO_BC |
|---|---|---|---|---|---|
| simple | **4.55** | 3.49 | 2.79 | 4.36 | 3.75 |
| random1 | **3.42** | 1.58 | 0.72 | 2.70 | 2.29 |
| random3 | **1.19** | 0.49 | 0.03 | 0.33 | 1.17 |
| unident_s | **5.27** | 3.62 | 1.84 | 2.51 | 2.58 |
| random0 | 0.37 | 0.49 | 0.00 | 0.21 | **1.44** |

When paired with the PPO_BC partner, we get the following average scores:

| layout | CoMeDi | XP | ADAP | SP | PBT |
|--------|--------|------|------|------|------|
| simple | **4.91** | 4.46 | 2.77 | 4.51 | 3.75 |
| random1 | **3.43** | 1.81 | 1.10 | 2.38 | 2.29 |
| random3 | **1.91** | 1.1 | 0.03 | 1.13 | 1.17 |
| unident_s | 2.94 | 3.16 | 2.59 | **4.52** | 2.58 |
| random0 | **1.56** | 0.88 | 0 | 0.25 | 1.44 |

From these results, we see that CoMeDi generally works well with unseen partners. It even surpasses the PPO_BC agent under most layouts, which uses human data in its training process. Interestingly, the results are inconsistent in the unident_s and random0 environments. When analyzing the trajectories of the best-performing agents in these cases, it seems like the conventions used by the partners just coincidentally aligned with the other agents.

When evaluating under self-play, we get the following average scores:

| layout | CoMeDi | XP | ADAP | SP | PPO_BC | PBT |
|--------|--------|------|------|------|--------|------|
| simple | **5.52** | 4.68 | 2.75 | 4.36 | 4.83 | 4.93 |
| random1 | **5.36** | 3.06 | 1.90 | 3.47 | 2.81 | 3.04 |
| random3 | 3.83 | 1.13 | 0.0 | **5.72** | 2.20 | 3.99 |
| unident_s | **8.44** | 4.40 | 4.42 | 7.84 | 2.95 | 4.33 |
| random0 | 4.67 | 2.95 | 0.0 | **5.98** | 2.06 | 4.30 |

These results generally are what we would expect. A pure self-play policy that does not need to consider diversity will likely have a strong score, which is why we sometimes find it scoring very high relative to the other baselines. However, CoMeDi often has a higher self-play score, because the additional diversity factor caused later conventions to be better, like in the Blind Bandits toy example.

## C Related Work

Concurrent to our work, methods for training diverse agents with respect to reward have been proposed.

In LIPO [7], they also consider cross-play as a diversity metric, but do not address the critical issue of handshakes that arise when minimizing cross-play. Their results reinforce our observations when the mixed-play weight, $\beta$, is set to 0. However, as we have noticed in our Balance Beam simulation results and the Overcooked user study, mixed-play has a large impact on creating good-faith conventions.

The ADVERSITY [8] work considers a zero-shot coordination framework in the game of Hanabi. They propose a belief reinterpretation model to address a similar "sabotaging" behavior that we experienced. This model is designed to tackle the issue of handshakes by finding a plausible distribution of self-play states that would result in the same observation received from cross-play and use this while training so agents cannot discriminate between self-play and cross-play observations. However, it is unclear whether belief reinterpretation would help in the games we examine in this paper since cross-play can potentially encounter observations that are completely impossible under self-play. Specifically, in Overcooked, all agents have access to the state from the prior frame, but conventions exist as a way to manage the workload of tasks between partners. In Hanabi, a core part of the game is predicting the underlying state of the game given the observation. As such, conventions are based around communicating information about the state implicitly through actions.

ADVERSITY uses the fact that multiple trajectories can generate the same action-observation history, which enables it to gain very strong results in Hanabi. However, this technique would fail in our settings because there are observations that are only possible under cross-play and not self-play, so belief reinterpretation would not be helpful. Therefore, although ADVERSITY and CoMeDi both attempt to address the problem of handshakes, their core assumptions are entirely different.

Since CoMeDi does not perform explicit belief reinterpretation, it will not be competitive with ADVERSITY on Hanabi, but it would still be able to train a sequence of agents.

In games where implicit communication to predict the underlying state is the core task, ADVERSITY is a strong choice for training a diverse set of agents. However, CoMeDi would be more effective in tasks where a team needs to divide a workload or commit to a particular strategy for effective coordination. We believe that these scenarios are more similar to the tasks that one would encounter in a robotics domain or typical video game setting.

## D  Limitations

Although our technique of creating a convention-aware agent using CoMeDi was able to surpass human-level performance in Overcooked, this technique has some drawbacks. Since the policy has no memory, training a convention-aware agent on a diverse set may lead to the AI breaking conventions established with humans earlier in the game, which is why one user reported that the CoMeDi agent sometimes breaks the established flow in Coordination Ring. Also, the BC-based algorithm for generating the convention-aware agent is often unable to account for human suboptimalities or transitions between conventions. For instance, some agents in Cramped Room would pick up an onion and expect the human to pick up a plate. If the human does not comply, it simply stands around instead of dropping off the onion and picking up a plate itself, because this type of action is never experienced under self-play for any convention.

On a theoretical level, CoMeDi does not provide any guarantees regarding the quality of agents. This is not unique to our algorithm, as statistical diversity techniques and reward shaping often changes the environment to the extent that "equilibrium conventions" are no longer guaranteed. Also, our solution to handshakes, mixed-play, implicitly assumes that re-establishing handshakes will come at some expense to the self-play scores. If this assumption is violated, as is the case with cheap-talk signals that aren't necessary for coordination, handshakes can be re-established at every timestep, effectively bypassing the mixed-play optimization. The issue of cheap-talk is a very tricky case in general when attempting to define diversity or robustness, because signals have no implicit meaning. In these cases, environment designers can remove extraneous cheap-talk signals or add nonuniform cost to communication, which has been effective in the realm of zero-shot communication [4].

## E  Broader Impact

We believe that CoMeDi can have a positive impact on game design and human-AI interaction in general. Being able to generate diverse conventions can allow game designers to understand the different strategies that players might try to use before extensive play-testing. Effective zero-shot coordination techniques would also help reduce the risk of misaligned conventions. We observe that, with proper tuning of the mixed-play weight, the convention-aware agent trained with CoMeDi learns to follow the lead of the human player, as indicated by the "followed user" section of the user survey. This is important for safety-critical applications like human-robot interaction tasks, because an overly assertive robot could unintentionally harm a human.

As a tool, it can directly be used for harmful ends, such as making it easier to cheat in multi-player games or generally conduct harm on others. Another potential effect of developing effective artificial zero-shot collaborators is that it could lead to more social withdrawal. In particular, if people who play video games start to strongly prefer playing with super-human AI collaborators over other humans, we may see people play less games with other people, which could counteract the prosocial benefits of cooperative gaming [12]. We therefore urge potential game designers and publishers who want to use CoMeDi to generate AI partners to evaluate the impact that artificial agents would have on their community.

## F  Creative Assets

Custom assets for this paper were digitally created by the authors without the assistance of AI image generation models. Stylistic inspiration was drawn from the Overcooked figures in [5] under the MIT license.

