# OpenReview forum: "Diverse Conventions for Human-AI Collaboration"
_NeurIPS.cc/2023/Conference — NeurIPS 2023 poster_

### Official Review · Reviewer_qsFc · 2023-06-29

**Soundness:** 2 fair
**Presentation:** 3 good
**Contribution:** 2 fair
**Rating:** 6
**Confidence:** 3

**Summary:**

This work suggests employing cross-play rewards to improve the measure of diversity in human-agent collaboration tasks, and presents an innovative training paradigm, namely, mixed-play, to address the challenge during optimizing cross-play, referred to as the "handshake" by the authors. As a result, the developed agent CoMeDi demonstrated superior performance in comparison to other baseline agents when collaborating with humans in the OverCooked game.

**Strengths:**

1. This paper is well-written and readable, and provides detailed appendices and codes to supplement the paper's algorithms, experiments, and implementation details.

2. This work tried to tackle an important problem domain - ad-hoc team play in human-agent collaboration research.

3. The experimental section fully validated the problems and necessity of self-play, cross-play, and mixed-play. The results of user studies on overcooked show the effectiveness of the proposed CoMeDi approach.

**Weaknesses:**

1. The behavior of human players tends to be non-stationary and noisy-rationally[1]. I am particularly interested in how the CoMiDi agent reacts to a player exhibiting random actions (such as a novice), or how the proposed CoMiDi method resolves this issue at the algorithmic level. This is because, upon experiencing the CoMiDi agent on the webpage (random1, random3, and scenario1_s Layout), I found that when I either refrain from any operation or deliberately obstruct one side of the path, the agent appears to be unintelligent and would not go around from the other side. Such performance contradicts the authors' claim that the CoMiDi agent surpasses human-level performance. Does this imply a lack of diversity in training or is it due to insufficient training?

2. The experimental section seems to resemble an ablation study process, with an absence of comparisons to state-of-the-art (SOTA) approaches, such as FCP[2], HSP[3]. While the authors assert that these SOTA methods are not drop-in replacements for self-play, I believe that the FCP and HSP approaches are not fundamentally distinct from SP, and they can be considered as improved methods of SP. I recommend that the authors conduct comparisons with these methods to make the proposed methods more solid.

3. The experimental section only discusses the results of partnering with expert humans. I am curious whether the authors have evaluated the CoMiDi agent's performance in collaboration with humans of varying skill levels (such as novices). Moreover, do all human partners prefer to collaborate with the CoMiDi agent over other agents? I believe user studies are crucial in the human-agent collaboration domain. Consequently, I suggest that the authors allocate more space in the main text to discuss user studies.

4. It is commendable that the author discusses the scenario where the number of teams exceeds two in the appendix. I am particularly intrigued by how the CoMiDi approach or the convention-based method can be extended to more intricate tasks, such as Google Football. It would be appreciated if the authors could offer insight in this respect.

[1] Anirudha Majumdar, Sumeet Singh, et al. Risk-sensitive inverse reinforcement learning via coherent risk models. 2017.

[2] DJ Strouse, Kevin R. McKee, et al. Collaborating with Humans without Human Data. 2021.

[3] Chao Yu, Jiaoxuan Gao, et al. Learning zero-shot cooperation with humans, assuming humans are biased. 2023.

**Questions:**

1. Why the CoMiDi agent is not as intelligent as expected for some relatively simple behaviors?

2. How the proposed method performs compared to the SOTA methods on OverCooked?

3. How does the CoMiDi agent perform in teams with different levels of humans?

4. What is the training manner of the proposed method, is it sequence training or joint training?

5. Did the author encounter any challenges other than Handshakes during training?

**Limitations:**

As discussed above, this work lacks the comparison of SOTA methods, and the user studies are not sufficient, which would limit the reliability of the proposed method. Besides, the authors should have further discussion on the scalability of the proposed method, such as in more complex environments.

---

> ### Author Rebuttal · Authors · 2023-08-10
>
> Thank you for the thorough review of our paper!
>
> We hope that the responses below answer your questions about our technique.
>
> ## Explanation for unintelligent behavior in simple scenarios
>
> Please refer to the “Clarification on the convention-aware agent” section in the common response above, where we clarify the intentions of the convention-aware agents.
>
> There are two main reasons for the bad behavior when interacting with CoMeDi: (1) the policy model and (2) the choice of QDagger.
>
> In our experiments, the policy network has no memory or recurrence, like the original Overcooked-AI paper; it simply chooses the action given the current observation. Therefore, if the partner chooses to stand still, the policy has no way of knowing how long the partner has been staying still. Correcting this behavior would result in a policy that randomly chooses to switch directions even when the partner is working correctly but slowly.
>
> The choice of QDagger means that we are simply cloning the self-play behavior of the diverse set of conventions. This means that policies implicitly assume that partners will choose an optimal action at each timestep, which is why it seems overly patient with an adversarial or novice partner.
>
> ## Comparison to FCP and HSP
>
> We believe that both FCP and HSP have valuable contributions to solving the challenge of Human-AI collaboration in Overcooked, but their techniques are largely orthogonal to our main goal. In our work, we specifically aim to generate a diverse set of conventions and we use QDagger in our evaluation as the standard technique for creating a convention-aware agent.
>
> In the Limitations and Future Work section of FCP, they write, “Our method currently relies on the manual process of initially training and selecting a diverse set of partners. This is not only time consuming, but also prone to researcher biases that may negatively influence the behavior of the created agent.” Similarly, in the Limitations and Future Work section of HSP, they write, “HSP requires domain knowledge to design a suitable set of events. [...]  So a future direction is to utilize learning-based methods to design rewards automatically.” Therefore, in terms of our goal of generating a diverse set of conventions, FCP requires manual selection and HSP requires domain knowledge to shape rewards (similar to other approaches to diversity that we discuss in section 2.3 and appendix B.1).
>
> However, we believe that using a technique like FCP could help create a better convention-aware agent than the QDagger algorithm, though we do not follow this direction in our experiments since it makes the answer to our central research question ambiguous.
>
> ## Clarification on Human Experiments
>
> We would like to clarify that our experiments when testing our AI agents were not with “expert humans” but paid participants with varying prior experiences in Overcooked, recruited through Prolific. There may be self-selection bias given the title of the study and description, but none of the participants have played our exact version of Overcooked prior to the study. However, the human-human studies had a single delegated expert human play with other participants who had varying levels of experience with Overcooked.
>
> Also, not all partners prefer to collaborate with CoMeDi. In the Coordination Ring, 21 out of 25 users put it as their top choice (4 of which were tied with another agent). In the Cramped Room, 17 out of 25 users put it as their top choice (4 of which were tied with another agent).
>
> ## Scalability to more Complex Environments
>
> When scaling to more complex environments, CoMeDi should be able to generate a diverse set of conventions if hyperparameters are well chosen for the first self-play convention. In particular, the choice of cross-play over statistical diversity means that our diversity metric will not be “tricked” by larger state spaces, and the choice of mixed-play means that our analysis of handshakes will still carry over to complex environments.
>
> However, one challenge of complex environments is sparse rewards. In particular, if rewards are very sparse, then cross-play minimization does not have a strong signal, so measuring diversity is more difficult. Also, complex environments may have low-cost handshakes, which is a problem described in the second paragraph of appendix section D.
>
> ## Sequence vs Joint Training
>
> In appendix section A.3, we explain why we chose sequential over joint training for the results in this paper. Sequential training ensured that we would not need to retrain from scratch if we wanted to increase the number of agents. However, it is possible to modify the algorithm to train jointly. Specifically, we can train all n conventions in parallel, but still ensure that each convention minimizes cross-play with the conventions of earlier indices to get a similar “cascading” effect present in sequential training.
>
> ## Challenges other than Handshakes
>
> When developing the CoMeDi algorithm, we learnt that creating separate self-play, cross-play, and mixed-play value networks was very important for stability. Although in theory these can be combined, we find that the high variance of using a single network resulted in significant training instability.

---

> > ### Comment · Reviewer_qsFc · 2023-08-17
> >
> > I appreciate the authors' clarification, which has addressed most of my concerns. I will increase my score.

---

### Official Review · Reviewer_Dzgz · 2023-07-01

**Soundness:** 3 good
**Presentation:** 3 good
**Contribution:** 3 good
**Rating:** 7
**Confidence:** 3

**Summary:**

The paper introduces a new approach for generating diverse conventions by using a training objective that balances increasing self-play performance and minimizing cross-play performance with previously discovered conventions. To avoid handshake mechanisms that can game the proposed objective, the technique also involves a mixed-play bonus/penalty, that leads the agents to be uncertain about whether they're currently playing in self or cross play. The advantages of the proposed method in recovering meaningfully distinct conventions (relative to prior work) are shown experimentally and with a user study.

**Strengths:**

### Originality and significance

Although this is not my main area of research, this approach seems novel and highly relevant to the community. There seems to be a lot of interest in conventions, especially in the context of human-AI collaboration. Given the empirical performance of the proposed method relative to prior approaches, this seems like a significant contribution to the subfield. The theoretical motivation to the method is an extra nice feature of the approach.

### Quality and Clarity

I found the approach as quite a clever way of choosing the space of possible conventions to consider (by restricting to ones that maximize self-play reward but minimize cross-play reward), avoiding the problems of assuming adversarial conventions. The concept of handshakes was well explained, and the idea of mixed-play seems like a good (although a little "hacky") solution.

The related work section was done well and the overall motivation for the paper was very well executed.

Overall, the paper was well-written and easy to read.

**Weaknesses:**

- The authors do not compare to behavior cloning play (i.e. "human-aware RL" from [4]) or similar approaches because it assumes access to expensive human data. I think it would still be very interesting to make this kind of comparison, because one long-standing challenge is to be able to effectively collaborate with humans without human data [26]. It would be interesting to see how far diversity approaches such as the one presented can get us. In other words, this could even enable to answer interesting questions such as "how much actual human data are these diversity approaches worth?". This seems especially interesting given the qualitative user reports.
- Rules of balance beam are pretty unclear – it might be worth trying to clarify them further
- A relevant recent work that the authors might not be aware of is: Who Needs to Know? Minimal Knowledge for Optimal Coordination [Lauffer, 2023]

**Questions:**

- There's one aspect of the paper that I don't understand (and might be worth clarifying more somewhere – unless I missed it). It seems like the whole paper is about generating diverse conventions, and so is the proposed method (e.g. in line 274 one tries to generate $\pi_n$). However, at evaluation time (e.g. in Overcooked) one must only choose one policy for CoMeDi. How is that done? In the conclusion, you say "we use a simple BC-based algorithm for training a convention-aware agent", but I don't see this referenced elsewhere?

Minor typos:
- line 129 spacing between commas. Footnotes have a space before preceding periods or even come before period (e.g. 140, 96)

**Limitations:**

With real humans, there are cognitive biases that might have them deviate from optimality arbitrarily, leading them to conventions that wouldn't be discovered by CoMeDi. Will this method will only work when humans are close enough to optimal? It might be worth acknowledging this as a limitation.

---

> ### Author Rebuttal · Authors · 2023-08-10
>
> Thank you for the thorough review of our paper!
>
> We hope that the responses below answer your questions about our technique.
>
> ## Choosing Convention-Aware Agent
>
> We explain the generation of a convention-aware agent in Equation 2, within section 2.1 of the paper, but we will edit the paper to re-emphasize this point during the Overcooked section. We also provide a more detailed explanation of this choice in the Common Response above.
>
> ## Comparison to BC-Play
>
> In the Common Response above, we compare the BC-Play (labeled PPO_BC) to our techniques when testing against PBT. Interestingly, we find that CoMeDi outperforms BC-Play by significant margins in simple, random1, and unident_s, and CoMeDi slightly outperforms BC-Play in random3. However, BC-Play does significantly better than CoMeDi at random0. In the original paper, we see that both BC-Play and PBT do very poorly when working directly with humans in the random0 layout, so it is unclear whether the issue is CoMeDi’s policy or PBT’s inflexibility.
>
> ## Rules of Balance Beam
>
> We go into more detail on the description of Balance Beam in appendix section B.6, but we can incorporate some more information into the main text. If you have any additional questions regarding the Balance Beam environment, we would be happy to answer them during the discussion period.
>
> ## Minimal Knowledge for Optimal Coordination
>
> Thank you for sharing this paper! The idea of needing prior knowledge to resolve strategic ambiguity is very important for Human-AI interaction. Their experiment on the modified version of Coordination Ring seems to indicate that despite the existence of various strategies (what we would call conventions), there exists a single best-response policy (VSEC) that can effectively determine which convention to follow just from the current state. We believe that the ideas of this paper can be helpful in determining how to construct a strong best response, and our technique for generating diverse conventions could help when trying to approximate their approach on more complex environments.
>
> ## Optimality of Humans
>
> In our Overcooked experiments, we work with real users who often deviate from optimal policies, and our convention-aware agent was able to handle these well. Since our policy is not conditioned on history, the diversity of our set of conventions is enough to ensure sufficient state coverage to minimize out-of-distribution behavior. Furthermore, the convention-aware policy implicitly acts with the assumption that their partner will take the optimal action at the next timestep, which causes it to be an “optimistic” policy that is forgiving to non-stationary and noisy-rational human behavior.
>
> As reviewer qsFc mentions in their testing, the optimistic behavior of QDagger may not be good when working with an uncooperative partner who intentionally takes no actions or blocks the other agents. However, alternatives to QDagger for using the diverse set of agents may be able to address more significant deviations from optimality.

---

> > ### Comment · Reviewer_Dzgz · 2023-08-13
> > **Response to rebuttal**
> >
> > Thank you so much for your response, and additional experiments.
> >
> > I found the comparisons with BC-Play particularly interesting!
> >
> > I do think that having more information about the Beam environment in the main text might be useful for readers, as I feel like it is essential to understand that environment's results. Also, thanks for the pointer to equation 2, I had missed or forgotten about it after seeing it – that might be worthwhile to emphasize with a heading or bolded paragraph title.
> >
> > Even after reading other reviews, I still stand by my score.

---

### Official Review · Reviewer_ghEw · 2023-07-03

**Soundness:** 3 good
**Presentation:** 3 good
**Contribution:** 3 good
**Rating:** 7
**Confidence:** 5

**Summary:**

The authors develop CoMeDi which optimizes a cross-play objective but avoids handshakes that can prevent cooperation using mixed play. The algorithm is evaluated in three environments of increasing complexity and on an Overcooked level with real human users.

**Strengths:**

The paper is written well and the ideas are well motivated both by filling gaps in the literature and by tackling key challenges that the authors introduce using conceptual examples as well as toy experiments. The quantitative and qualitative evaluation on Overcooked with real human participants is the gold-standard for studying Coordination. Furthermore, the CoMeDi algorithm is an elegant solution to the problem of generating diverse conventions and cleanly avoids many of the pitfalls that can arise from adversarial training. The authors have released a playable version of their agent to the web.

**Weaknesses:**

The Overcooked analysis was only done with a single grid out of many. This leaves open the possibility that the algorithm doesn’t work well on the more varied set of environments that are typically studied. I would raise my score if the algorithm could be evaluated on the full Overcooked suite (either with people or with baseline agents). This is critical because otherwise, the results might have arisen from a cherry-picked experiment.

No formal analysis of the algorithm — how does this approach scale when the number of possible conventions or agents grows?

Little evaluation done on the Overcooked environment. Besides overall reward, how well do these agents fare? How far off are the cross-play and self-play rewards from the results with people? How many conventions were generated for the Overcooked environment?

**Questions:**

We need more information about the overcooked results. How well does the final policy perform in simulation against other policies. Not just in terms of human performance but self-play & cross-play with the other computational agents?

This is a fast moving space, could the authors devote some words to how this approach differs from Adversarial Diversity in Hanabi or Generating diverse cooperative agents by learning incompatible policies. The latter seems very similar in approach.

Were the experiments with people done under time pressure? I noticed when playing that game online that time ticked independently from my own moves. How does that differ from prior work on human evaluation?

What does it mean to over-correcting conventions during mixed-play? What are the implications? How carefully do these parameters need to be balanced? Can the same hyper-parameters be used across all Overcooked scenarios?

**Limitations:**

Yes

---

> ### Author Rebuttal · Authors · 2023-08-10
>
> Thank you for the thorough review of our paper! We evaluated our algorithm across the full Overcooked suite with baseline agents, as you recommended. We believe that some of your additional questions have been answered in the appendix (within the supplementary materials as Diverse_Conventions_Appendix.pdf), but we expand on each of your questions below.
>
> ## Additional Overcooked Results + Details
>
> We present the full suite of tests against baseline agents in the common rebuttal above. We also recommend looking at section B.8 in the appendix for more information about the generation of Overcooked agents in the Cramped Room and Coordination Ring environments. For ADAP, XP, and CoMeDi, we generate a population of 8 conventions. If you have other results you would like to see, we would be happy to provide them during the discussion period.
>
> ## Scalability of Approach
>
> In section A.5 of the appendix, we describe an approach for extending our algorithm beyond two players. However, we believe that deciding how exactly to collect the cross-play buffers in large team settings is still an open research question.
>
> When there are a large number of conventions, CoMeDi still greedily chooses the least compatible undiscovered convention, so it can handle these situations in theory. However, one practical issue is that the number of cross-play buffers and cross-play value functions needed grows linearly with the number of conventions, so over time CoMeDi becomes significantly slower than pure self-play. A possible alternative is to have multiple independent generations of CoMeDi, which may cover a wide range of conventions without consuming as much extra time or space, but this may result in a loss of diversity since independent invocations could rediscover similar conventions.
>
> ## Comparison to ADVERSITY and LIPO
>
> In section C of the appendix, we have a detailed comparison of CoMeDi to LIPO and ADVERSITY. At a high level, LIPO is similar to our baseline of pure cross-play with no usage of mixed-play. On the other hand, ADVERSITY proposes a solution to “sabotaging” behavior, similar to our description of handshakes, but their technique of belief reinterpretation is specifically designed for games where partial observability is a major component of working with conventions. In particular, ADVERSITY would not work in Balance Beam or Overcooked since these are fully observable. CoMeDi does not explicitly perform belief reinterpretation, so it will likely not be as strong as ADVERSITY in Hanabi.
>
> ## Time Pressure
>
> We use a similar experimental setup to the one used in the original Overcooked paper, but we reduce the time to 40 seconds. Overcooked requires that players wait for soup to finish cooking, so time has to tick independently from moves, which is common to all Overcooked human-AI experiments as far as we know.
>
> ## Overcorrecting for Handshakes in Mixed-Play
>
> In Balance Beam, we observe that when the mixed-play weight, beta, is set too high, conventions trained with CoMeDi learn to match the behavior of the original (pure MAPPO) convention. In general, having too high of a weight for mixed-play incentivizes agents to eliminate handshakes along with genuine differences in conventions. In practice, we find that choosing alpha=0.5 and beta=1.0 gives good results, and we only use these hyperparameters across all Overcooked environments.

---

> > ### Comment · Reviewer_ghEw · 2023-08-15
> >
> > Thank you for the detailed response. I have raised my score and recommend the paper be accepted

---

### Official Review · Reviewer_9YDu · 2023-07-05

**Soundness:** 3 good
**Presentation:** 3 good
**Contribution:** 3 good
**Rating:** 7
**Confidence:** 3

**Summary:**

This paper introduces a new approach to dealing with poor generalisation performance in human-AI settings when actors may have developed different coordination conventions.

**Strengths:**

In general, I think the work is original and very clear in its presentation. In addition, the paper attempts to challenge a very obvious and significant issue in the domain of human-AI collaboration and takes a well-formulated approach to the challenge. More specifically:

- I think the problem is well formulated, and in particular the related work in relation to diversity approaches in RL is strong.
- I found the method generally easy to follow, and enjoyed its presentation in terms of creating all of the building blocks towards the final algorithm.
- On the presented empirical evidence the results are strong.

**Weaknesses:**

Whilst the empirical evidence presented is strong, my concerns with this paper are with the extent of the results and how they actually go about answering the question of this paper. Therefore, my main concern is as follows:

- The paper argues that their approach is able to handle an arbitrarily diverse set of other agent conventions. Whilst this may be true, I am unconvinced by the empirical results that may try to back this up. In my opinion, the problem with the experiments that were picked is that there just really do not exist that diverse a pool of conventions in the first place. In particular, in the Overcooked environment I would be amazed if any human player undertook a strategy that was not immediately obvious. Therefore, I would have been very interested in seeing some form of analysis that was able to show that the algorithm was being put up against a diverse set of conventions in its test set.

Note, I have played with the demo of the paper and I do think that it appears to be a strong approach even under strange conventions. However, my own testing is probably not a strong enough scientific argument to back this up. I'm happy to bump my score up for the paper if the authors can convince me that the above is not as much of an issue that I believe it might be.

**Questions:**

- The fact that there are two tunable parameters here $\alpha$ and $\beta$ could potentially become a difficult balancing act in other environments. Could the authors comment a little more on how they would suggest selecting these parameters outside of full on hyper parameter tuning? Or is this the only viable option?



**Limitations:**

I would have been interested to see where the algorithm may feel and if the authors could have stress tested it a little more than in the paper.

---

> ### Author Rebuttal · Authors · 2023-08-10
>
> Thank you for the thorough review of our paper! We’re glad that you played with the demo and think that our approach works well, even under strange conventions.
>
> We hope that the responses below answer your questions about our technique.
>
> ## Diversity of Overcooked’s Human test set
>
> We agree that most conventions that humans would naturally converge upon feel obvious on the Overcooked test suite. In the Coordination ring specifically, most differences in conventions are based on the direction of motion, i.e. clockwise versus counterclockwise. Humans also tend to be able to adapt to conventions easily, so determining whether CoMeDi adapts to human’s conventions instead of requiring humans to adapt is unclear from purely looking at average scores.
>
> From the user survey results (Figure 7 of main text), we see that CoMeDi has a higher score for following the user’s lead, indicating that it is able to follow the conventions of users. Additionally, we see that an understanding of conventions has a large impact on final scores since the only difference between convention-aware agents is the set of self-play rollouts used for QDagger. Although human conventions seem very obvious to us, the failures of SP, ADAP, and XP indicate that these conventions are harder for the policy networks to understand.
>
> We also ran a new experiment where we explicitly predict the probability of being in a CoMeDi convention conditioned on the observation of the current player. Since our usage of QDagger can be approximated as a gated mixture-of-experts model, we can explicitly construct a “gate” network by training a simple classifier that determines which convention an observation belongs to. Out of the 5000 observations (from 25 players * 200 observations/player) that CoMeDi encountered when working with users in the Coordination Ring, 2349 of them were aligned with a specific convention with a probability over 50%, which we refer to as “convention-specific” observations. Of these 2349 observations, we can calculate the frequency of each convention being the most likely one in the table below:
>
> |	0 | 1 |	2 | 3 | 4 |	   5 |	6 |   7 |
> |---|---|---|---|---|---|---|---|
> | 0.50 | 0 | 0.25 | 0 | 0 | 0.05 | 0.04 | .15 |
>
> As you can see, convention 0 (the pure self-play policy) behaves well in around 50% of the convention-dependent states, but knowledge of the other states is still vital. In particular, we see that convention 2 and convention 7 also seem to align well with human conventions. For most players, convention 0 is the most likely convention, but this is not universal. For one player, 65% of the convention-specific observations aligned with convention 7 while only 27% aligned with convention 0. For another player, 44% of the convention-specific observations aligned with convention 2 while 39% aligned with convention 0. Overall, if we consider which convention each player followed a plurality of the time, 20 players followed convention 0, 3 players followed convention 7, and 2 players followed convention 2. However, there was never a case when the convention-aware agent only used information from a single convention, indicating that the understanding of multiple conventions made the policy more robust overall.
>
> We also ran this experiment on the Cramped Room results, and we found that 2124 observations were aligned to a specific convention with probability over 50%. We show the frequency of each convention being the most likely one below:
>
> |	0 |	1 | 2 |	3 |	4 |	5 |	6 |   7 |
> |---|---|---|---|---|---|---|---|
> | 0.05 | 0.25 | 0 | 0.01 | 0.45 | 0.11 | 0.04 | .09 |
>
> When considering which convention each player followed a plurality of the time, 16 players followed convention 4, 8 players followed convention 1, and 1 player followed convention 5.
>
> Tying back to your original question, within the scope of Overcooked we see non-trivial evidence of diversity among the human conventions.
>
> ## How to select hyperparameters
>
> We detailed some guidelines for hyperparameter tuning in the appendix section A.4. This process involves keeping the standard hyperparameters from self-play, and tuning the cross-play weight before tuning the mixed-play weight. In practice, we find that CoMeDi is pretty robust to hyperparameters and very often gives reasonable policies with a cross-play weight of 0.5 and a mixed-play weight of 1 (or 0 when handshakes are impossible).
>
> ## Failure Modes of CoMeDi
>
> In appendix section D, we examine some cases where CoMeDi will likely struggle. One issue is working with environments with cheap-talk channels or handshakes that are extremely easy to execute, because mixed-play only helps when executing handshakes negatively impacts the reward.

---

> > ### Comment · Reviewer_9YDu · 2023-08-16
> >
> > Thank you for taking the time to address my concerns. I am happy to increase my score as I believe my main concern is addressed through the new 'diversity' experiments. I would expect to see this included in the manuscript in some capacity.

---

### Author Rebuttal · Authors · 2023-08-10

We thank all the reviewers for their thoughtful feedback. We appreciate that all reviewers found the paper well-written while tackling key challenges in human-AI collaboration research. We are happy that reviewers found our algorithm “well-formulated” (9YDu) and “elegant” (ghEw) while “avoiding the problems of assuming adversarial conventions” (Dzgz) and that our experimental section “fully validated the problems and necessity of self-play, cross-play, and mixed-play.” (qsFc)

The reviewers asked a wide range of questions, which we respond to in detail in their individual responses. For the rest of the common response, we clarify the purpose of the convention-aware agent, highlight some new simulation experiments on the rest of the Overcooked suite, and present some minor corrections.

## Clarification on the convention-aware agent:

We define the process of creating a convention-aware agent in equation 2 of the problem definition section. This is essentially the QDagger algorithm, but instead of using expert human generated trajectories we treat the diverse set of conventions as the “experts” and collect their self-play rollouts. We can approximately treat this as a distilled mixture-of-experts model, where each “expert” is a generated convention, and they are gated by the probability of experiencing an observation in its self-play rollout.

We chose this definition of a “convention-aware agent” because it is a good proxy for answering our research question of whether a set of conventions is **diverse** and **high-quality**. For instance, if the set is not diverse but has high quality, this convention-aware agent will have a large space of out-of-distribution observations, leading to poor performance on the test set. If the set is diverse but has low quality, the agent will consistently take bad actions even when its observations are in-distribution. A set that is both diverse and high-quality would have a convention-aware agent that consistently takes good actions since its space of out-of-distribution observations is small.

Other techniques for generating an agent using a set of conventions, typically based around generating a common best response agent, do not serve as good proxies for answering our question due to the more complex dynamics involved with this process. In particular, a common best response agent to a set of very low quality partners will learn to never rely on their partner, resulting in an agent that never defers work to their partner but can get strong scores in environments where it can act alone. On the other hand, a common best response agent paired with a diverse set of extremely high quality partners may become too reliant on their partner, so it would not work well when partnered with novices.

## Experiments on Full Overcooked Suite:

Our human-AI experiments only covered the Coordination Ring (random1) and Cramped Room (simple) environments, because we considered them to be the canonical Overcooked layouts. However, since reviewer ghEw asked us to evaluate our algorithms on the full Overcooked suite against baseline agents, we trained models for the 3 other layouts used in the original Overcooked-AI paper: Counter Circuit (random3), Asymmetric Advantages (unident_s), and Forced Coordination (random0). To train the new models, we use the same hyperparameters as the other Overcooked models. We were unable to use the original BC or human-proxy models from the original Overcooked-AI work due to technical difficulties when working with outdated versions of python libraries, but we were able to use the tensorflow-js versions of their PPO_BC and PBT models.

When paired with the PBT partner, we get the following average scores:

| layout	| CoMeDi |   XP | ADAP |   SP | PPO_BC |
|---|---|---|---|---|---|
| simple	|   **4.55** | 3.49 | 2.79 | 4.36 |   3.75 |
| random1   |   **3.42** | 1.58 | 0.72 | 2.70 |   2.29 |
| random3   |   **1.19** | 0.49 | 0.03 | 0.33 |   1.17 |
| unident_s |   **5.27** | 3.62 | 1.84 | 2.51 |   2.58 |
| random0   |   0.37 | 0.49 | 0.00 | 0.21 |   **1.44** |

When paired with the PPO_BC partner, we get the following average scores:

| layout	| CoMeDi |   XP | ADAP |   SP | PBT |
|---|---|---|---|---|---|
| simple	|   **4.91** | 4.46 | 2.77 | 4.51 | 	3.75 |
| random1   |   **3.43** | 1.81 | 1.10 | 2.38 | 	2.29 |
| random3   |   **1.91** |  1.1 | 0.03 | 1.13 | 	1.17 |
| unident_s |   2.94 | 3.16 | 2.59 | **4.52** | 	2.58 |
| random0   |   **1.56** | 0.88 |	0 | 0.25 | 	1.44 |

From these results, we see that CoMeDi generally works well with unseen partners! It even surpasses the PPO_BC agent under most layouts, which uses human data in its training process. Interestingly, the results are inconsistent in the unident_s and random0 environments. When analyzing the trajectories of the best-performing agents in these cases, it seems like the conventions used by the partners just coincidentally aligned with the other agents.

When evaluating under self-play, we get the following average scores:

| layout	| CoMeDi |   XP | ADAP |   SP | PPO_BC |  PBT |
|---|---|---|---|---|---|---|
| simple	|   **5.52** | 4.68 | 2.75 | 4.36 |   4.83 | 4.93 |
| random1   |   **5.36** | 3.06 | 1.90 | 3.47 |   2.81 | 3.04 |
| random3   |   3.83 | 1.13 |  0.0 | **5.72** |   2.20 | 3.99 |
| unident_s |   **8.44** | 4.40 | 4.42 | 7.84 |   2.95 | 4.33 |
| random0   |   4.67 | 2.95 |  0.0 | **5.98** |   2.06 | 4.30 |

These results generally are what we would expect. A pure self-play policy that does not need to consider diversity will likely have a strong score, which is why we sometimes find it scoring very high relative to the other baselines. However, CoMeDi often has a higher self-play score, because the additional diversity factor caused later conventions to be better, like in the Blind Bandits toy example.

## Minor Corrections:

We will fix the formatting issues brought up by Dzgz. We will also correct equation 2 by deleting the extra $\pi(a|o)$ term.

---

### Decision · Program_Chairs · 2023-09-21

**Decision:**

Accept (poster)

**Comment:**

The authors study human-AI coordination in fully observable settings.
Specifically, they train one agent as a best response to a pool of diverse agents, as is common practice in literature.
They also build on adversarial cross play as an established method for driving meaningful diversity in multi-agent populations.

The key contribution is a thorough experimental evaluation of a simplified version of ADVERSITY, adapted to the fully observable setting.
Like adversity, the partner is either chosen to be cooperative or adversarial with a given probability to avoid "sabotaging" behaviour.

I recommend this paper for acceptance, under the following conditions:
1) "Our key insight is to instead use the incompatibility of different conventions as a measure of diversity" -> This claim needs to be removed form the paper, since both ADVERSITY and LIPO already use this mechanism for diversity.

2) For the purpose of academic integrity, the section on "Handshakes: Challenges of Optimizing Cross-Play" needs to be rewritten to properly consider prior work (ADVERSITY) which extensively introduces (and solves) this problem.

3) The paper introduces Dec-POMDPs in the background section but only tests on fully observable environments. Claiming that the method is applicable to Dec-POMDPs is thus misleading. This is further supported by the paper using pi(a|o) in the paper, rather than conditioning on action-observation histories (tau) as necessary under partial observability.

4) The authors should explicitly compare and contrast to ADVERSITY and LIPO in the main body of the paper. ADVERSITY in particular contains an opponent mixing step which closely mirrors the mixed-play component of this work. Based on this, authors should also adapt their claims regarding the novelty of the mixed play algorithm.